# Interplay of diruthenium catalyst in controlling enantioselective propargylic substitution reactions with visible light-generated alkyl radicals

Yulin Zhang[1], Yoshiaki Tanabe [1] ✉, Shogo Kuriyama [1], Ken Sakata [2] ✉ & Yoshiaki Nishibayashi [1] ✉

Transition metal-catalyzed enantioselective free radical substitution reactions have recently attracted attention as convenient and important building tools in synthetic chemistry, although construction of stereogenic carbon centers at the propargylic position of propargylic alcohols by reactions with free radicals remains unchallenged. Here we present a strategy to control enantioselective propargylic substitution reactions with alkyl radicals under photoredox conditions by applying dual photoredox and diruthenium catalytic system, where the photoredox catalyst generates alkyl radicals from 4-alkyl-1,4-dihydropyridines, and the diruthenium core with a chiral ligand traps propargylic alcohols and alkyl radicals to guide enantioselective alkylation at the propargylic position, leading to high yields of propargylic alkylated products containing a quaternary stereogenic carbon center at the propargylic position with a high enantioselectivity. The result described in this paper provides the successful example of transition metal-catalyzed enantioselective propargylic substitution reactions with free alkyl radicals.

Recently, transition metal-catalyzed enantioselective free radical substitution reactions have gaining attention as versatile synthetic tools, for free radical substitution reactions are more compatible with many functional groups, and tedious protection/deprotection steps can be rather avoided compared to those in ionic substitution reactions such as nucleophilic substitution reactions[1-3]. In addition, radicals are less reactive to Brønsted or Lewis acids, which tend to be required to facilitate dissociation of leaving groups for ionic substitution reactions[3]. Therefore, free radical substitution reactions provide an attractive approach to deal with substrate limitations for ionic substitution reactions. For instance, transition metal-catalyzed enantioselective allylic substitution reactions with soft nucleophiles ($pK_a < 25$) have been well established as convenient methods to introduce stereogenic carbon centers adjacent to alkene moieties (Fig. 1a)[4,5], although reactions with hard nucleophiles ($pK_a > 25$) including alkyl nucleophiles, not activated by functional groups such as methyl derivatives attached with ketone, aldehyde, or ether groups, have been found to be difficult due to the requirement of harsh conditions[6]. This problem has been very recently shown to be overcome by using free alkyl radicals as the alternative to nucleophiles (Fig. 1b), where the radicals are prepared in situ by the employment of photoredox catalysis[7-10].

Similarly to the transition metal-catalyzed enantioselective allylic substitution reactions, transition metal-catalyzed enantioselective propargylic substitution reactions with soft nucleophiles have been very recently developed to introduce stereogenic carbon centers adjacent to alkyne moieties by using ruthenium-[11-13], copper-[14-20], or nickel-based[21-23]

[1]Department of Applied Chemistry, School of Engineering, The University of Tokyo, Hongo, Bunkyo-ku, Tokyo 113-8656, Japan. [2]Faculty of Pharmaceutical Sciences, Toho University, Miyama, Funabashi, Chiba 274-8510, Japan. ✉e-mail: ytanabe@g.ecc.u-tokyo.ac.jp; ken.sakata@phar.toho-u.ac.jp; ynishiba@g.ecc.u-tokyo.ac.jp

Fig. 1 | Strategies for transition metal-catalyzed enantioselective substitution reactions at carbon centers adjacent to alkene or alkyne moieties. a Transition metal-catalyzed enantioselective allylic substitution reactions on treatment with nucleophiles. b Transition metal-catalyzed enantioselective allylic substitution reactions on treatment with free radicals. c Transition metal-catalyzed enantioselective propargylic substitution reactions on treatment with nucleophiles or free radicals. d This work: enantioselective propargylic substitution reactions catalyzed by dual photoredox- and diruthenium system. DC distinct catalyst, LG leaving group, NuH nucleophile, PC photocatalyst.

catalysts (Fig. 1c)[24–26]. However, in contrast to the allylic substitution reactions, application of free radical reactions to the transition metal-catalyzed enantioselective substitution reactions at the propargylic positions of propargylic alcohols or their derivatives has not yet explored, for appropriate methods to control the generation and reactivity of free radicals including inhibition of direct reactions of radicals with alkyne moieties are required. To overcome this problem, we have focused upon the use of two reaction systems: (1) the use of 4-alkyl-1,4-dihydropyridines[27–30] as free radical sources under photoredox conditions, and (2) the use of optically active thiolate-bridged diruthenium complexes [{Cp*RuCl(μ-SR*)}2] (Cp* = η⁵-C5Me5, R* = chiral substituent)[31–34] as catalysts for enantioselective propargylic substitution reactions (Fig. 1d). Here, 4-alkyl-1,4-dihydropyridines have been shown to supply alkyl radicals to afford alkylated products under photoredox reaction conditions, providing a platform of alkylation reagents designed for the use under milder reaction conditions[27–29]. Application of 4-alkyl-1,4-dihydropyridines can be expanded by dual photoredox- and transition metal-catalyzed systems[30,35,36] toward simple alkylation[7,8,37,38] and alkylative reactions[9,39–43]. Regarding the thiolate-bridged diruthenium complexes, which do not catalyze alkylative reactions of alkyne moieties with radicals as known for nickel-based catalysts[41–43], one ruthenium atom of the diruthenium core works as an

electron reservoir for the other substrate-converting ruthenium atom via the electronic communication between the two metals, expected to stabilize not only coordinatively unsaturated species but also radical species[31,32]. Indeed, reactive intermediates derived from thiolate-bridged diruthenium complexes were shown to capture free radicals[44]. In total, the strategy to operate enantioselective propargylic alkylation of propargylic alcohols can be summarized as shown in Fig. 1d, where the iridium-based photoredox catalyst generates alkyl radicals from 4-alkyl-1,4-dihydropyridines[29,30], and the diruthenium core catalyzes propargylic substitution reactions retaining alkyne moieties unaffected by radicals[31,32].

Here, we show the preparation of propargylic alkylated products with a chiral carbon center at the propargylic position via the enantioselective propargylic substitution reaction with free radicals, where the reactivity of radicals is well governed by the dual photoredox- and diruthenium-catalyzed system, providing the successful example of enantioselective propargylic substitution reaction with free radicals.

## Results
### Reaction optimization
As the beginning, we carried out reactions of 1-phenyl-prop-2-yn-1-ols with substituents introduced at the propargylic position

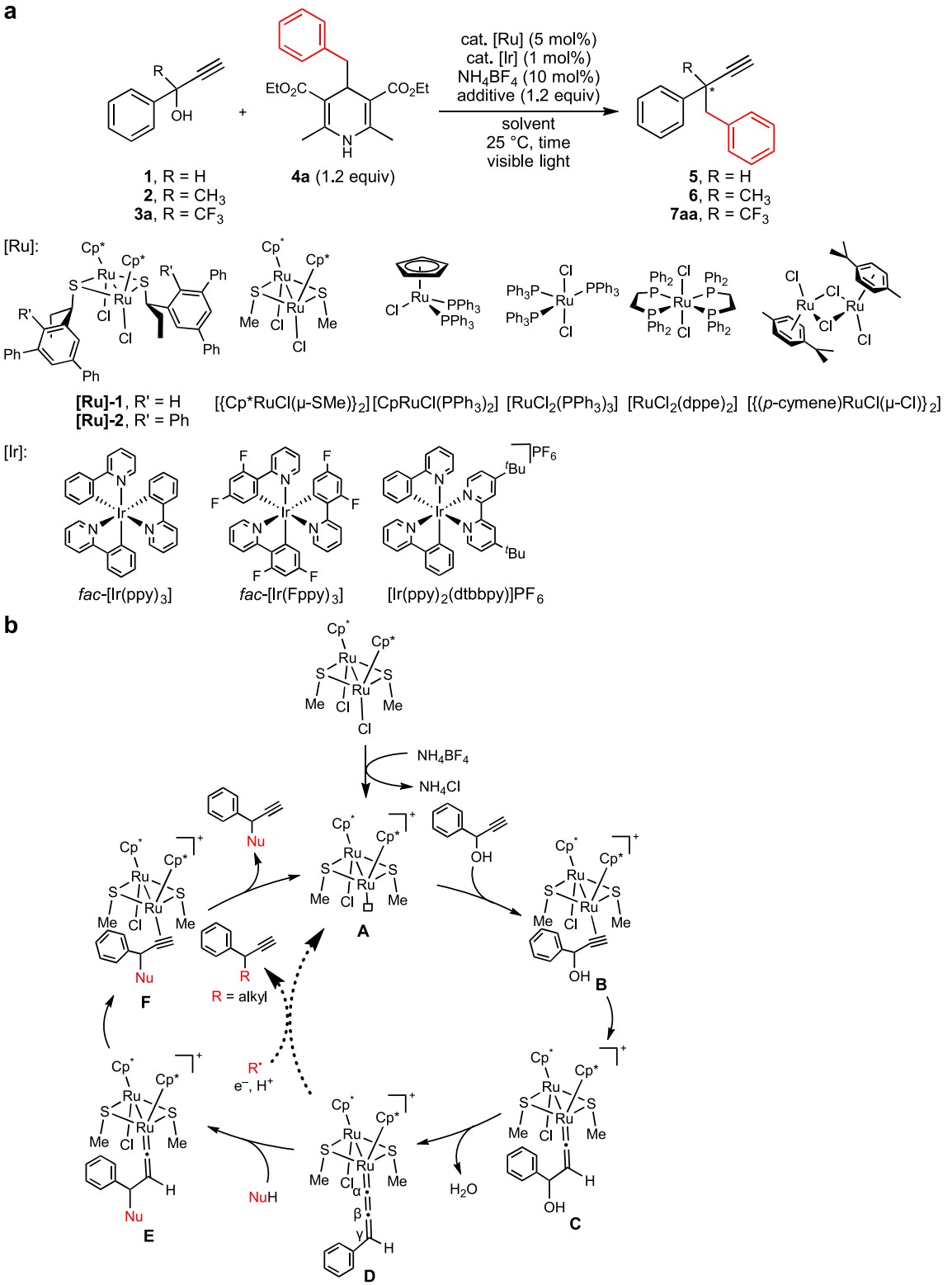

**Fig. 2 | Photoredox- and diruthenium-catalyzed enantioselective propargylic alkylation. a** Screening of substrates and optimization of the reaction conditions. [Ir], iridium-based photocatalyst; [Ru], ruthenium-based catalyst for propargylic substitution reactions. **b** Plausible reaction pathways for propargylic substitution reactions of propargylic alcohols with a nucleophile or an alkyl radical catalyzed by a thiolate-bridged diruthenium complex. NuH, nucleophile.

(HC≡CC(OH)(R)Ph, R = H (**1**), Me (**2**), or CF$_3$ (**3a**)) with 1.2 equiv. of 4-benzyl-1,4-dihydropyridine (**4a**) in the presence of 5 mol% of an optically active thiolate-bridged diruthenium complex [{Cp*RuCl(μ-SR*)}$_2$] (**[Ru]−1**, SR* = (R)-SCH(Et)C$_6$H$_3$Ph$_2$−3,5)[11], 1 mol% of *fac*-[Ir(ppy)$_3$] (ppy = 2-(pyridyl)phenyl), and 10 mol% of NH$_4$BF$_4$ in

1,2-dichloroethane (ClCH$_2$CH$_2$Cl) at 25 °C for 18 h under visible light irradiation (Fig. 2a and Table 1, entries 1–6). The reaction of **1** with 1.2 equiv. of **4a** afforded the corresponding propargylic alkylated product (**5**) in 42% yield with 29% ee (Table 1, entry 1). Interestingly, addition of 1.2 equiv. of BF$_3$·Et$_2$O increased both the yield and enantioselectivity of

**Table 1 | Screening of substrates and optimization of the reaction conditions for the photoredox- and diruthenium-catalyzed enantioselective propargylic alkylation[a]**

| Entry | Alcohol | [Ru] | [Ir] | Additive | Solvent | Time (h) | Product | Yield (%)[b] | ee (%) |
|---|---|---|---|---|---|---|---|---|---|
| 1 | **1** | **[Ru]-1** | *fac*-[Ir(ppy)$_3$] | none | ClCH$_2$CH$_2$Cl | 18 | **5** | 42 | 29 |
| 2 | **1** | **[Ru]-1** | *fac*-[Ir(ppy)$_3$] | BF$_3$·Et$_2$O | ClCH$_2$CH$_2$Cl | 18 | **5** | 73 | 48 |
| 3 | **2** | **[Ru]-1** | *fac*-[Ir(ppy)$_3$] | none | ClCH$_2$CH$_2$Cl | 18 | **6** | trace | – |
| 4 | **2** | **[Ru]-1** | *fac*-[Ir(ppy)$_3$] | BF$_3$·Et$_2$O | ClCH$_2$CH$_2$Cl | 18 | **6** | trace | – |
| 5 | **3a** | **[Ru]-1** | *fac*-[Ir(ppy)$_3$] | none | ClCH$_2$CH$_2$Cl | 18 | **7aa** | trace | – |
| 6 | **3a** | **[Ru]-1** | *fac*-[Ir(ppy)$_3$] | BF$_3$·Et$_2$O | ClCH$_2$CH$_2$Cl | 18 | **7aa** | 47 | 86 |
| 7 | **3a** | **[Ru]-2** | *fac*-[Ir(ppy)$_3$] | BF$_3$·Et$_2$O | ClCH$_2$CH$_2$Cl | 18 | **7aa** | 35 | 93 |
| 8 | **3a** | [{Cp*RuCl(μ-SMe)}$_2$] | *fac*-[Ir(ppy)$_3$] | BF$_3$·Et$_2$O | ClCH$_2$CH$_2$Cl | 18 | **7aa** | 84 | – |
| 9 | **3a** | [CpRuCl(PPh$_3$)$_2$] | *fac*-[Ir(ppy)$_3$] | BF$_3$·Et$_2$O | ClCH$_2$CH$_2$Cl | 18 | **7aa** | trace | – |
| 10 | **3a** | [RuCl$_2$(PPh$_3$)$_3$] | *fac*-[Ir(ppy)$_3$] | BF$_3$·Et$_2$O | ClCH$_2$CH$_2$Cl | 18 | **7aa** | n.d. | – |
| 11 | **3a** | [RuCl$_2$(dppe)$_2$] | *fac*-[Ir(ppy)$_3$] | BF$_3$·Et$_2$O | ClCH$_2$CH$_2$Cl | 18 | **7aa** | n.d. | – |
| 12 | **3a** | [{(*p*-cymene)RuCl(μ-Cl)}$_2$] | *fac*-[Ir(ppy)$_3$] | BF$_3$·Et$_2$O | ClCH$_2$CH$_2$Cl | 18 | **7aa** | n.d. | – |
| 13 | **3a** | **[Ru]-2** | *fac*-[Ir(ppy)$_3$] | BF$_3$·Et$_2$O | ClCH$_2$CH$_2$Cl | 48 | **7aa** | 81 | 94 |
| 14 | **3a** | **[Ru]-2** | *fac*-[Ir(Fppy)$_3$] | BF$_3$·Et$_2$O | ClCH$_2$CH$_2$Cl | 48 | **7aa** | 54 | 51 |
| 15 | **3a** | **[Ru]-2** | [Ir(ppy)$_2$(dtbbpy)]PF$_6$ | BF$_3$·Et$_2$O | ClCH$_2$CH$_2$Cl | 48 | **7aa** | 39 | 73 |
| 16 | **3a** | **[Ru]-2** | *fac*-[Ir(ppy)$_3$] | BF$_3$·Et$_2$O | CH$_2$Cl$_2$ | 48 | **7aa** | trace | – |
| 17 | **3a** | **[Ru]-2** | *fac*-[Ir(ppy)$_3$] | BF$_3$·Et$_2$O | Cl(CH$_2$)$_3$Cl | 48 | **7aa** | 32 | 83 |
| 18 | **3a** | **[Ru]-2** | *fac*-[Ir(ppy)$_3$] | B(OH)$_3$ | ClCH$_2$CH$_2$Cl | 48 | **7aa** | trace | – |
| 19 | **3a** | **[Ru]-2** | *fac*-[Ir(ppy)$_3$] | Sc(OTf)$_3$ | ClCH$_2$CH$_2$Cl | 48 | **7aa** | trace | – |
| 20 | **3a** | **[Ru]-2** | *fac*-[Ir(ppy)$_3$] | BCl$_3$ | ClCH$_2$CH$_2$Cl | 48 | **7aa** | trace | – |
| 21[c] | **3a** | **[Ru]-2** | *fac*-[Ir(ppy)$_3$] | BF$_3$·Et$_2$O | ClCH$_2$CH$_2$Cl | 48 | **7aa** | 13 | 93 |
| 22[d] | **3a** | **[Ru]-2** | *fac*-[Ir(ppy)$_3$] | BF$_3$·Et$_2$O | ClCH$_2$CH$_2$Cl | 48 | **7aa** | trace | – |
| 23[e] | **3a** | **[Ru]-2** | *fac*-[Ir(ppy)$_3$] | none | ClCH$_2$CH$_2$Cl | 48 | **7aa** | trace | – |
| 24[c,e] | **3a** | **[Ru]-2** | *fac*-[Ir(ppy)$_3$] | BF$_3$·Et$_2$O | ClCH$_2$CH$_2$Cl | 48 | **7aa** | 15 | 92 |
| 25 | **3a** | **[Ru]-2** | none | BF$_3$·Et$_2$O | ClCH$_2$CH$_2$Cl | 48 | **7aa** | trace | – |
| 26[f] | **3a** | **[Ru]-2** | *fac*-[Ir(ppy)$_3$] | BF$_3$·Et$_2$O | ClCH$_2$CH$_2$Cl | 48 | **7aa** | trace | – |
| 27 | **3a** | none | *fac*-[Ir(ppy)$_3$] | BF$_3$·Et$_2$O | ClCH$_2$CH$_2$Cl | 48 | **7aa** | n.d. | – |

[a]Reaction conditions (Fig. 2a): **1**, **2**, or **3a** (0.10 mmol), **4a** (0.12 mmol), [Ru] (0.005 mmol), [Ir] (0.001 mmol), NH$_4$BF$_4$ (0.01 mmol), additive (0.12 mmol), and solvent (2.0 mL) at 25 °C for time under visible light irradiation with a 12 W white LED. [Ru], ruthenium-based catalyst for propargylic substitution reactions. [Ir], iridium-based photocatalyst.
[b]Isolated yield of the products (**5**, **6**, or **7aa**). trace, detected by GC-MS for crude mixture but not isolable. n.d., not detected by GC-MS.
[c]10 mol% instead of 1.2 equiv. of BF$_3$·Et$_2$O.
[d]H$_2$O (0.12 mmol) added.
[e]Anhydrous MgSO$_4$ (0.20 mmol) added.
[f]Without visible light irradiation.

**5** up to 73 and 48% ee, respectively (Table 1, entry 2). In contrast, the reaction of **2**, a tertiary propargylic alcohol bearing a methyl group introduced at the propargylic position, with 1.2 equiv. of **4a** gave only a trace amount of the desired alkylated product (**6**) under similar reaction conditions even in the presence of 1.2 equiv. of BF$_3$·Et$_2$O (Table 1, entries 3 and 4). The reaction of **3a**, another tertiary propargylic alcohol bearing a trifluoromethyl group at the propargylic position, with 1.2 equiv. of **4a** gave only a trace amount of the desired alkylated product (**7aa**) (Table 1, entry 5). However, addition of 1.2 equiv. of BF$_3$·Et$_2$O fairly increased the yield of **7aa** up to 47% with a high enantioselectivity such as 86% ee (Table 1, entry 6). These experimental results indicate that introduction of a trifluoromethyl group at the propargylic position is an essential factor to achieve the high enantioselectivity of the propargylic alkylated products.

Next, we examined varieties of catalysts for the propargylic substitution reaction (Table 1, entries 6–12). When the reaction of **3a** with 1.2 equiv. of **4a** was carried out in the presence of 5 mol% of another optically active thiolate-bridged diruthenium complex [{Cp*RuCl(μ-SR*)}$_2$] (**[Ru]−2**, SR* = (*R*)-SCH(Et)C$_6$H$_2$Ph$_3$–2,3,5)[11] instead of **[Ru]−1** as a ruthenium catalyst, isolated yield of **7aa** was decreased to 35%, but the enantioselectivity was fairly increased to 93% ee (Table 1, entry 7). Interestingly, yield of **7aa**, obtained as a mixture of racemates, increased up to 84% when [{Cp*RuCl(μ-SMe)}$_2$], another

thiolate-bridged diruthenium complex with non-chiral centers, was used as the ruthenium catalyst instead of **[Ru]−1** or **[Ru]−2** (Table 1, entry 8). In addition, several mononuclear and binuclear ruthenium complexes such as [CpRuCl(PPh$_3$)$_2$] (Cp = η$^5$-C$_5$H$_5$), [RuCl$_2$(PPh$_3$)$_3$], [RuCl$_2$(dppe)$_2$], and [{(*p*-cymene)RuCl(μ-Cl)}$_2$] were further examined as catalysts. However, only a trace amount of **7aa** was obtained when [CpRuCl(PPh$_3$)$_2$] was used as a catalyst (Table 1, entry 9), or **7aa** was not obtained at all when [RuCl$_2$(PPh$_3$)$_3$], [RuCl$_2$(dppe)$_2$], or [{(*p*-cymene)RuCl(μ-Cl)}$_2$] was used as a catalyst (Table 1, entries 10–12).

It must be noted that our group has utilized several thiolate-bridged diruthenium complexes as the catalysts for propargylic substitution reactions of propargylic alcohols with nucleophiles. Here, thiolate-bridged diruthenium core containing ruthenium–ruthenium bond has been shown to the key for the catalysis to proceed, where the ancillary ruthenium center of the diruthenium core works as an electron reservoir for the substrate-converting ruthenium center via the electronic communication between the two metals to stabilize reactive intermediates including a coordinatively unsaturated complex (**A**) obtained by the reaction of the corresponding precursory diruthenium complex ([{Cp*RuCl(μ-SMe)}$_2$] as a typical example) with NH$_4$BF$_4$, a π-alkyne complex (**B**) obtained via the coordination of a propargylic alcohol (1-phenylprop-2-yn-1-ol as a typical example), a vinylidene complex (**C**) obtained via the 1,2-migration of the acetylenic

proton, an allenylidene complex (**D**) obtained via the loss of $H_2O$, another vinylidene complex (**E**) obtained via the attack of a nucleophile at the γ-carbon atom, and another π-alkyne complex (**F**) obtained via the 1,2-migration of the β-hydrogen atom, which liberates the propargylic substituted product to regenerates the coordinatively unsaturated complex **A** (solid pathway in Fig. 2b)[31,32]. The propargylic substitution reactions of propargylic alcohols with alkyl radicals presented in this manuscript have been also shown to be identical to thiolate-bridged diruthenium complexes, and likely follow the reaction pathway similar to Fig. 2b with almost the same reaction pathway for the formation an of allenylidene complex (**D**) by the reaction of a coordinatively unsaturated species (**A**) with a propargylic alcohol, but with slightly different reaction pathway for the radical substitution reactions; i.e. capture of an alkyl radical, the addition of a proton, and a single electron transfer occurs against the allenylidene complex **D**, where radical redox-relay reactions are furnished by the diruthenium core working as an electron pool (dotted pathway in Fig. 2b).

It should be cautioned that starting materials (both **3a** and **4a**) were recovered for enantioselective propargylic alkylation employing either **[Ru]−1** or **[Ru]−2** as a ruthenium catalyst (Table 1, entries 6 and 7), thus elongation of the reaction time up to 48 h was examined. When the reaction of **3a** with 1.2 equiv. of **4a** was carried out in the presence of 5 mol% of **[Ru]−2**, 1 mol% of *fac*-[Ir(ppy)₃], 10 mol% of $NH_4BF_4$, and 1.2 equiv. of $BF_3 \cdot Et_2O$ in $ClCH_2CH_2Cl$ at 25 °C for 48 h under visible light irradiation, the yield of **7aa** was increased up to 81% with a slightly higher enantioselectivity at 94% ee (Table 1, entry 13). Use of other iridium-based photoredox catalysts such as *fac*-[Ir(Fppy)₃] (Fppy = 3,5-difluoro-2-(pyridyl)phenyl) and [Ir(ppy)₂(dtbbpy)]PF₆ (dtbbpy = 4,4′-di-*tert*-butyl-2,2′-bipyridine) instead of *fac*-[Ir(ppy)₃] was also examined, where yields of **7aa** became slightly lower with a lower enantioselectivity (Table 1, entries 14 and 15).

Next, we examined optimization of other reaction conditions (Table 1, entries 16–27). When reactions were carried out in other solvents such as $CH_2Cl_2$ and $Cl(CH_2)_3Cl$, yields of **7aa** got lower with a lower enantioselectivity (Table 1, entries 16 and 17) compared to those in $ClCH_2CH_2Cl$. Reactions using some other Lewis acids such as $B(OH)_3$, $Sc(OTf)_3$, and $BCl_3$ instead of $BF_3 \cdot Et_2O$ were also examined, but trace amounts of **7aa** detectable only by GC-MS were formed in all cases (Table 1, entries 18–20). In addition, reducing the amount of $BF_3 \cdot Et_2O$ down to 10 mol% did not change the enantioselectivity of **7aa** with 93% ee but decreased the isolated yield of **7aa** to 13% (Table 1, entry 21), demonstrating that the addition of almost a stoichiometric amount of $BF_3 \cdot Et_2O$ is necessary to enhance the reaction. It needs to be comprehended that the present substitution reaction is also the dehydration process, giving a stoichiometric amount of $H_2O$ as a byproduct. Indeed, the addition of 1.2 equiv. of $H_2O$ was shown to inhibit the reaction (Table 1, entry 22). However, the addition of 2.0 equiv. of anhydrous $MgSO_4$ in the absence of $BF_3 \cdot Et_2O$ did not give the desired product (Table 1, entry 23), while the addition of both 2.0 equiv. of anhydrous $MgSO_4$ and 10 mol% of $BF_3 \cdot Et_2O$ gave **7aa** in 15% yield with 92% ee (Table 1, entry 24), demonstrating that main roles of $BF_3 \cdot Et_2O$ may be not only to work as a drying reagent to trap $H_2O$, but also to work as an accelerator for the dehydration process more efficiently than other Lewis acids ($B(OH)_3$, $Sc(OTf)_3$, and $BCl_3$) and anhydrous $MgSO_4$. Finally, additional control experiments were investigated. In the absence of photoredox catalyst but under visible light irradiation (Table 1, entry 25) or in the presence of photoredox catalyst but under dark (Table 1, entry 26), only trace amounts of **7aa** were obtained, while formation of **7aa** was perfectly not observed in the absence of ruthenium catalyst (Table 1, entry 27), clarifying that the photoredox catalyst, light, and ruthenium catalyst are all necessary to carry out reactions.

It should be remarked that free alkyl radicals generated from 4-alkyl-1,4-dihydropyridines under photoredox conditions were already employed as alkylation reagents in the dual photoredox- and palladium-catalyzed propargylic substitution reactions of propargylic esters to afford the propargylic alkylated products, although enantioselectivity was not achieved at all[37]. In this reaction system, both photoredox and palladium catalysts work as photosensitizers under visible light irradiation, where a single electron transfer (SET) between the propargylic esters and excited palladium catalysts occurs to afford the propargylic radicals, which react with the alkyl radicals, generated from 4-alkyl-1,4-dihydropyridines via a SET process with photoredox catalysts, to afford the corresponding propargylic alkylated products. Furthermore, photoinduced enantioselective propargylic substitution reactions were only reported for propargylic cyanation catalyzed by dual photoredox- and copper-catalyzed system, where a SET between the propargylic esters and photoinduced photoredox catalysts occurs to afford the propargylic radicals, which react with the nucleophilic copper cyanate complexes, formed via reactions of chiral copper catalysts and CN⁻ in situ, to afford the corresponding propargylic cyanated products[45–47] or cyanated allenes[48]. Thus, the results described in this present manuscript are the successful example of enantioselective propargylic alkylation of propargylic alcohols with free radicals. Moreover, construction a quaternary stereogenic carbon center at the propargylic position was also achieved[49].

## Substrate scope and application

Taking the reaction conditions of entry 13 in Table 1 as optimized, reactions of **3a** with several 4-alkyl-1,4-dihydropyridines (**4**) to obtain the corresponding propargylic alkylated products were next examined (Fig. 3a). The use of 4-benzyl-1,4-dihydropyridine derivatives, where either electron-donating (**4b**, Me; **4c**, MeO; **4d**, OH) or electron-withdrawing substituents (**4e**, F; **4f**, Cl; **4g**, Br; **4h**, CF₃; **4i**, MeOC(=O)) were imported at the 4-position of the benzene ring, afforded the corresponding propargylic alkylated products in good to high yields with a high enantioselectivity (**7ab**, 78% yield, 94% ee; **7ac**, 79% yield, 92% ee; **7ad**, 70% yield, 92% ee; **7ae**, 80% yield, 92% ee; **7af**, 74% yield, 90% ee; **7ag**, 68% yield, 94% ee; **7ah**, 74% yield, 91% ee; **7ai**, 75% yield, 96% ee), demonstrating that even phenolic OH group is tolerant (**7ad**), while the highest enantioselectivity at 96% ee was achieved for that containing benzoate ester (**7ai**). On the other hand, the use of the 4-benzyl-1,4-dihydropyridine derivatives, where bulky substituents were imported at the 4-position of the benzene ring (**4j**, ᵗBu; **4k**, Ph), afforded the corresponding propargylic alkylated products in good yields with a little lower enantioselectivity (**7aj**, 70% yield, 84% ee; **7ak**, 71% yield, 83% ee). 4-Benzyl-1,4-dihydropyridine derivatives with either electron-donating substituents imported at the 3-position (**4l**, Me; **4m**, F; **4n**, Cl; **4o**, Br) or the 2-position (**4p**, Me; **4q**, F) of the benzene ring were also applicable to afford the corresponding propargylic alkylated products in good yields with a high enantioselectivity (**7al**, 73% yield, 90% ee; **7am**, 80% yield, 90% ee; **7an**, 77% yield, 91% ee; **7ao**, 72% yield, 93% ee; **7ap**, 74% yield, 90% ee; **7aq**, 78% yield, 93% ee).

Scope of alkyl radicals is not limited to substituted benzyl radicals. Indeed, 1,4-dihydropyridine derivatives with arylmethyl groups such as 1-naphthylmethyl (**4r**), 2-naphthylmethyl (**4s**), piperonyl (**4t**), 2-furylmethyl (**4u**), and 2-thienylmethyl (**4v**) substituents imported at the 4-position of the dihydropyridine skeleton could be successfully converted into the corresponding propargylic alkylated products in good to high yields with a high enantioselectivity (**7ar**, 69% yield, 91% ee; **7as**, 71% yield, 95% ee; **7at**, 66% yield, 91% ee; **7au**, 80% yield, 90% ee; **7av**, 75% yield, 94% ee), providing a synthetic tool to construct heterocycle-containing alkynes. On the other hand, the use of 1,4-dihydropyridine derivatives with a pyridine-, ether, or amine-containing methyl group, a secondary alkyl group, or a long chain primary alkyl group (R² = 3-pyridylmethyl, benzyloxymethyl, *N,N*-dibenzylaminomethyl-, 3-pentyl, or *n*-nonyl) imported at the 4-position of the dihydropyridine skeleton gave only trace amounts of desired products, leading to failure of their isolation (Fig. 3b), which might be due to either their steric bulkiness or instability

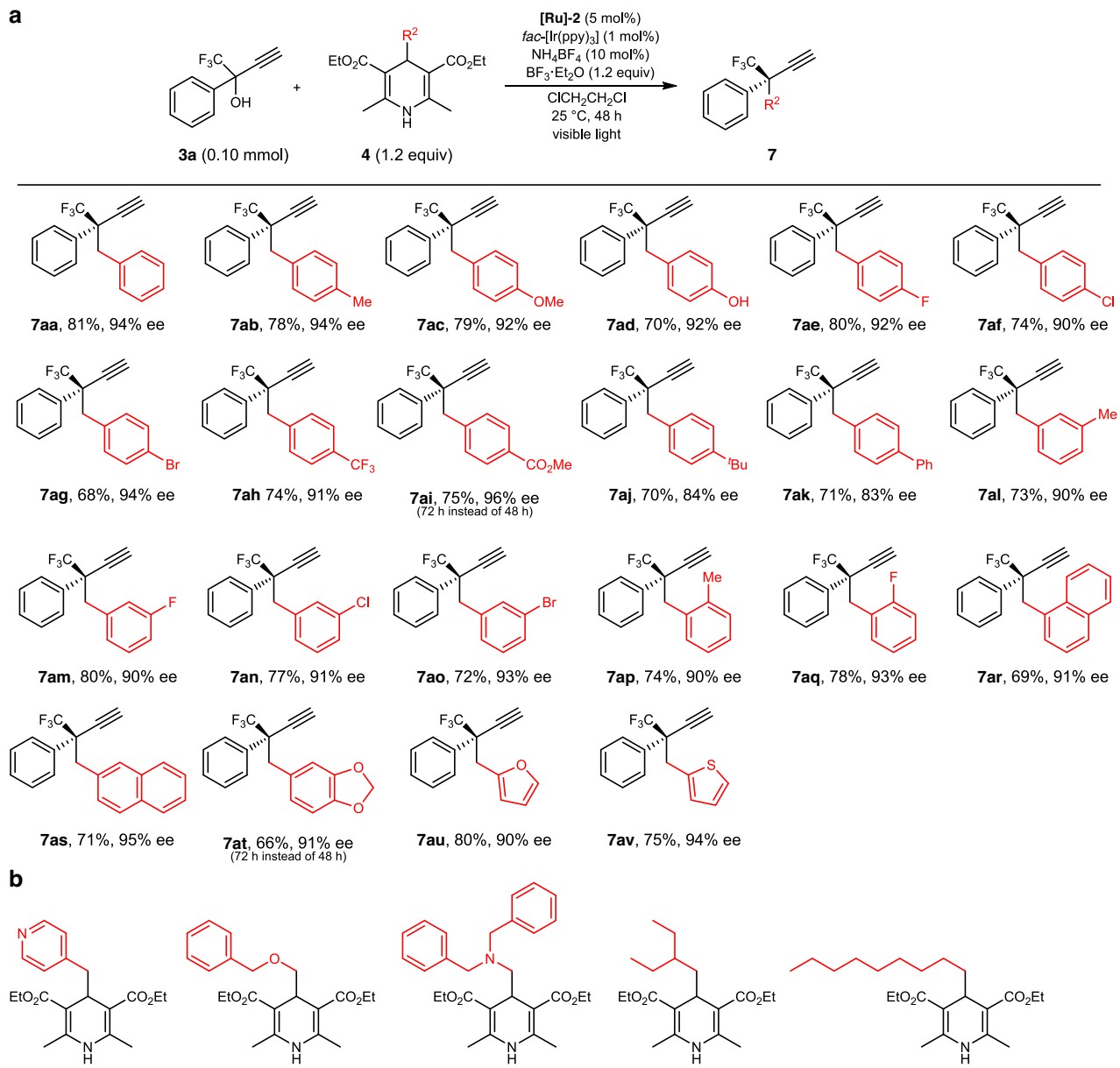

**Fig. 3 | Scope of 1,4-dihydropyridines for photoredox- and diruthenium-catalyzed enantioselective propargylic alkylation. a** Propargylic alkylated products with isolated yield. **b** 4-Alkyl-1,4-dihydropyridines with the formation of trace amounts of the desired products detected by GC-MS but not isolable for the reactions of **3a** with **4**.

of the corresponding alkyl radicals[50] generated under photoredox conditions.

Subsequently, propargylic substitution reactions of various propargylic alcohols (**3**) with **4a** were investigated (Fig. 4a). 1,1,1-Trifluoro-2-phenyl-but-3-yn-2-ol derivatives with either electron-donating (**3b**, Me; **3c**, Et; **3d**, MeO) or electron-withdrawing substituents (**3e**, F; **3f**, Cl; **3g**, Br; **3h**, CF₃) imported at the 4-position of the benzene ring were successfully alkylated to afford the desired products in good to high yields with a high enantioselectivity (**7ba**, 81% yield, 90% ee; **7ca**, 76% yield, 90% ee; **7da**, 71% yield, 91% ee; **7ea**, 77% yield, 94% ee; **7fa**, 74% yield, 90% ee; **7ga**, 70% yield, 92% ee; **7 ha**, 83% yield, 93% ee). 1,1,1-Trifluoro-2-phenyl-but-3-yn-2-ol derivatives, where either electron-donating or electron-withdrawing substituents were imported at the 3-position of the benzene ring (**3i**, Me; **3j**, F; **3k**, Cl; **3l**, Br), were also applicable to afford the corresponding propargylic alkylated products in good yields with a high enantioselectivity (**7ia**, 67% yield, 90% ee; **7ja**, 72% yield, 91% ee; **7ka**, 65% yield, 95% ee; **7la**, 70% yield, 90% ee).

Interestingly, not only propargylic alcohols containing phenyl derivatives (**3a**–**3l**) but also those containing heterocycles such as 2-furyl and 2-thienyl groups (**3m** and **3n**) were shown to be converted into the corresponding propargylic alkylated products in good yields with a high enantioselectivity (**7ma**, 76% yield, 96% ee; **7na**, 65% yield, 90% ee). However, the reaction of 1,1,1-trifluoro-2-phenyl-but-3-yn-2-ol derivatives (HC≡CC(OH)(CF₃)R¹) with substituents (Me or F) at the 2-position of the benzene ring (R¹ = 2-tolyl or 2-fluorophenyl) or 2-([1,1'-biphenyl]-4-yl)-1,1,1-trifluorobut-3-yn-2-ol (R¹ = 2-naphthyl) with **4a** gave only trace amounts of the desired products, presumably due to their steric hindrance (Fig. 4b). In addition, formation of the corresponding alkylated products was not observed at all by GC-MS for the reactions of 3-(trifluoromethyl)pent-1-yn-3-ol (HC≡CC(OH)(CF₃)Et) with **4a**, or of 1,1,1-trifluoro-2-phenylpent-3-yn-2-ol (MeC≡CC(OH)(CF₃)Ph) with **4a** (Fig. 4b). Namely, introduction of an aromatic moiety (such as substituted phenyl, 2-furyl, or 2-thienyl group) at the propargylic position of propargylic alcohols is necessary for the propargylic

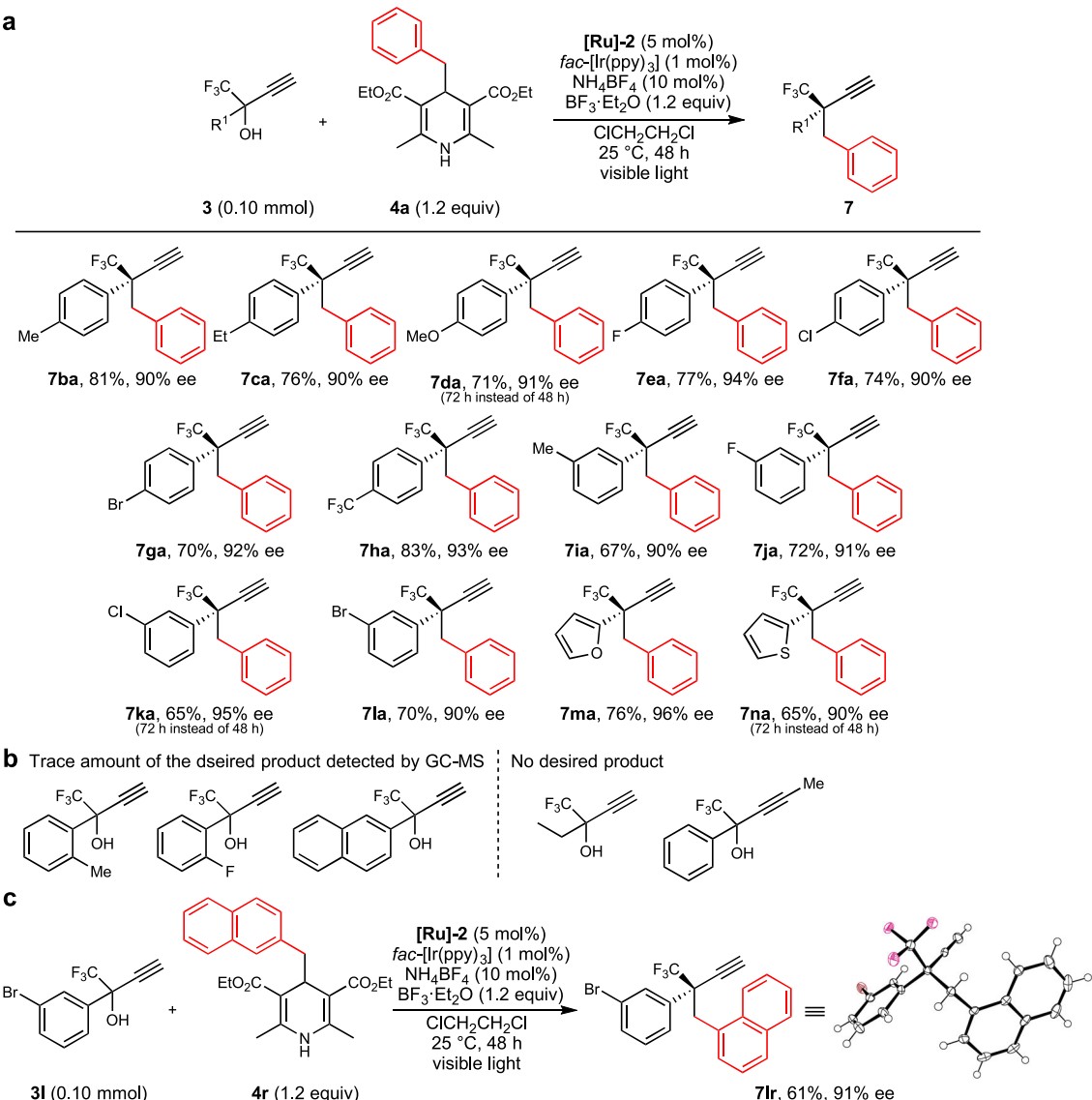

**Fig. 4 | Scope of propargylic alcohols for photoredox- and diruthenium-catalyzed enantioselective propargylic alkylation. a** Propargylic alkylated products with isolated yield. **b** Propargylic alcohols with the formation of trace amounts of the desired products detected by GC-MS but not isolable for the reactions of **3** with **4a**, or without the formation of the desired products for the reactions of **3** with **4a**. **c** Absolute configuration determination.

alkylation with 4-alkyl-1,4-dihydropyridines under photoredox conditions, while the propargylic moiety of the propargylic alcohols must contain terminal ethynyl group to form ruthenium–allenylidene intermediates.

The reaction of **3l** with **4r** was further examined to obtain the corresponding propargylic alkylated product (**7lr**) in 61% yield with a high enantioselectivity at 91% ee (Fig. 4c). The molecular structure of **7lr** has been determined after further purification with HPLC and recrystallization from Et$_2$O by a single-crystal X-ray diffraction study, which has clarified that the absolute configuration of **7lr** is (*R*), the chirality of which is same with those obtained for the ruthenium-catalyzed enantioselective propargylic phosphinylation of propargylic alcohols similarly bearing both CF$_3$ and aromatic groups at the propargylic position[13,34].

Interestingly, 1,1-difluoro-2-phenyl-but-3-yn-2-ol (**8**), where CF$_2$H group was introduced at the propargylic position instead of CF$_3$ group, was found to react with **4a** to afford the corresponding propargylic alkylated product (**9**) in 75% yield, although the enantioselectivity at 63% ee was significantly lower than that obtained for **7aa** at 94% ee (Fig. 5a), demonstrating that the existence of trifluoromethyl group at

the propargylic position plays a critical role for the enantioselective propargylic alkylation[34].

The practicality and synthetic potential of the enantioselective propargylic alkylation was then investigated. We first performed a larger-scale reaction of **3a** (1.00 mmol) with **4a** (1.20 mmol) under the optimized reaction conditions to obtain the propargylic alkylated product **7aa** in 75% isolated yield (0.75 mmol) with keeping a high enantioselectivity at 93% ee (Fig. 5b). Further transformations of **7aa** using this batch with an enantioselectivity at 93% ee was then examined, because terminal alkyne moieties have a wide range of synthetic utility[51]. Indeed, AgNO$_3$-catalyzed electrophilic bromination[52] of **7aa** with NBS (*N*-bromosuccinimide) afforded the corresponding bromoalkyne (**10**), while the Sonogashira coupling[53] of **7aa** with PhI afforded the corresponding internal alkyne (**11**), both in high yields without any loss of the optical purity of **7aa**, providing a synthetic tool for the enantioselective preparation of internal alkynes with propargylic alkyl substituents. Furthermore, Pd/C-catalyzed hydrogenation[54] of **7aa** gave the corresponding diphenylalkane (**12**) in 96% yield, while the photoredox- and NiCl$_2$·6H$_2$O-catalyzed hydroalkylation of **7aa** with **4a**, the synthetic method very recently reported by

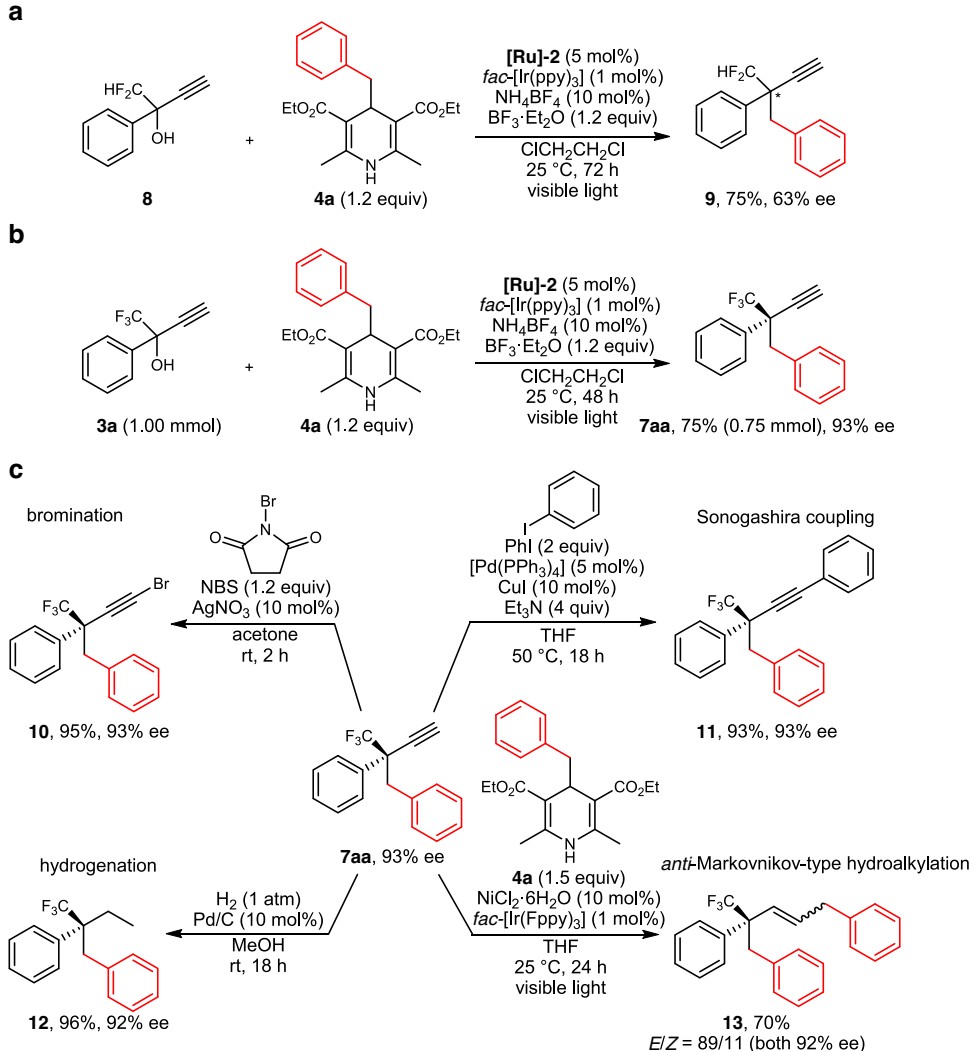

**Fig. 5 | Additional scope and application for enantioselective propargylic alkylation. a** Enantioselective propargylic alkylation of a propargylic alcohol bearing CF₂H instead of CF₃. **b** Large-scale preparation of a chiral propargylic alkylated product. **c** Transformations of a chiral propargylic alkylated product via silver-catalyzed electrophilic bromination with NBS, Sonogashira coupling with PhI, Pd/C-catalyzed hydrogenation with H₂, and photoredox- and nickel-catalyzed *anti*-Markovnikov-type hydroalkylation with 4-alkyl-1,4-dihydropyridine.

our group[42], afforded *anti*-Markovnikov-type alkylated alkene (**13**) in 70% yield as a mixture of (*E*)- and (*Z*)-isomers (*E/Z* = 89/11), all obtained almost without the loss of the optical purity of **7aa** (Fig. 5c).

## Mechanistic studies

To obtain more information on the enantioselective induction, we have conducted stoichiometric and catalytic reactions of the ruthenium–allenylidene complex [Cp*RuCl(μ-SR*)₂Ru{C=C=C(CF₃)Ph}Cp*]BF₄ (**14**, SR* = (*R*)-SCH(Et)C₆H₃Ph₂-3,5), which was originally prepared by the reaction of **[Ru]-1** with **3a** in the presence of NH₄BF₄ and anhydrous MgSO₄[13], and now has been shown to be prepared by the similar reaction in the presence of BF₃·Et₂O as a dehydrating reagent instead of anhydrous MgSO₄ (Fig. 6a). When the stoichiometric reaction of **14** with 1.2 equiv. of **4a** was carried out in the presence of 1 mol% of *fac*-[Ir(ppy)₃] and 1.0 equiv. of **3b** but in the absence of BF₃·Et₂O in ClCH₂CH₂Cl at 25 °C for 48 h under visible light irradiation, the propargylic alkylated product **7aa** was obtained in 66% yield with 88% ee (Fig. 6b). These results indicate that BF₃·Et₂O is not concerned with the alkylation of the allenylidene complex but likely with the dehydration process for the formation of allenylidene complex at least for propargylic alcohols containing CF₃ group at the propargylic position.

Separately, the reaction of **3a** with 1.2 equiv. of **4a** in the presence of 5 mol% of **14**, 1 mol% of *fac*-[Ir(ppy)₃], and 1.2 equiv. of BF₃·Et₂O in ClCH₂CH₂Cl at 25 °C for 48 h was examined to afford **7aa** in 85% yield with 85% ee (Fig. 6c). This result has clarified that ruthenium–allenylidene complexes such as **14** worked as possible key reactive intermediates for the propargylic alkylation as have been confirmed for the propargylic substitution reactions of propargylic alcohols with nucleophiles[12,13,55].

In order to get more mechanistic information of the reaction pathway, we next investigated the propargylic alkylation of **3a** with **4a** in the presence of 1.2 equiv. of (2,2,6,6-tetramethylpiperidin-1-yl)oxyl (TEMPO) to capture radicals under the optimized reaction conditions. As a result, formation of propargylic alkylation product **7aa** was perfectly inhibited, while 1-(benzyloxy)-2,2,6,6-tetramethylpiperidine (**15**)[56], the TEMPO-trapped benzyl adduct, was obtained in 34% yield (Fig. 6d). Formation of other TEMPO-trapped compounds was not observed from the reaction mixture. At present, we cannot exclude the possiblity of the formation of propargylic radicals as reactive intermediates[37,45–48,57–60], because we used tertiary propargylic alcohols bearing the CF₃ moiety as the substrate, where the corresponding propargylic radicals are difficult to be trapped by TEMPO.

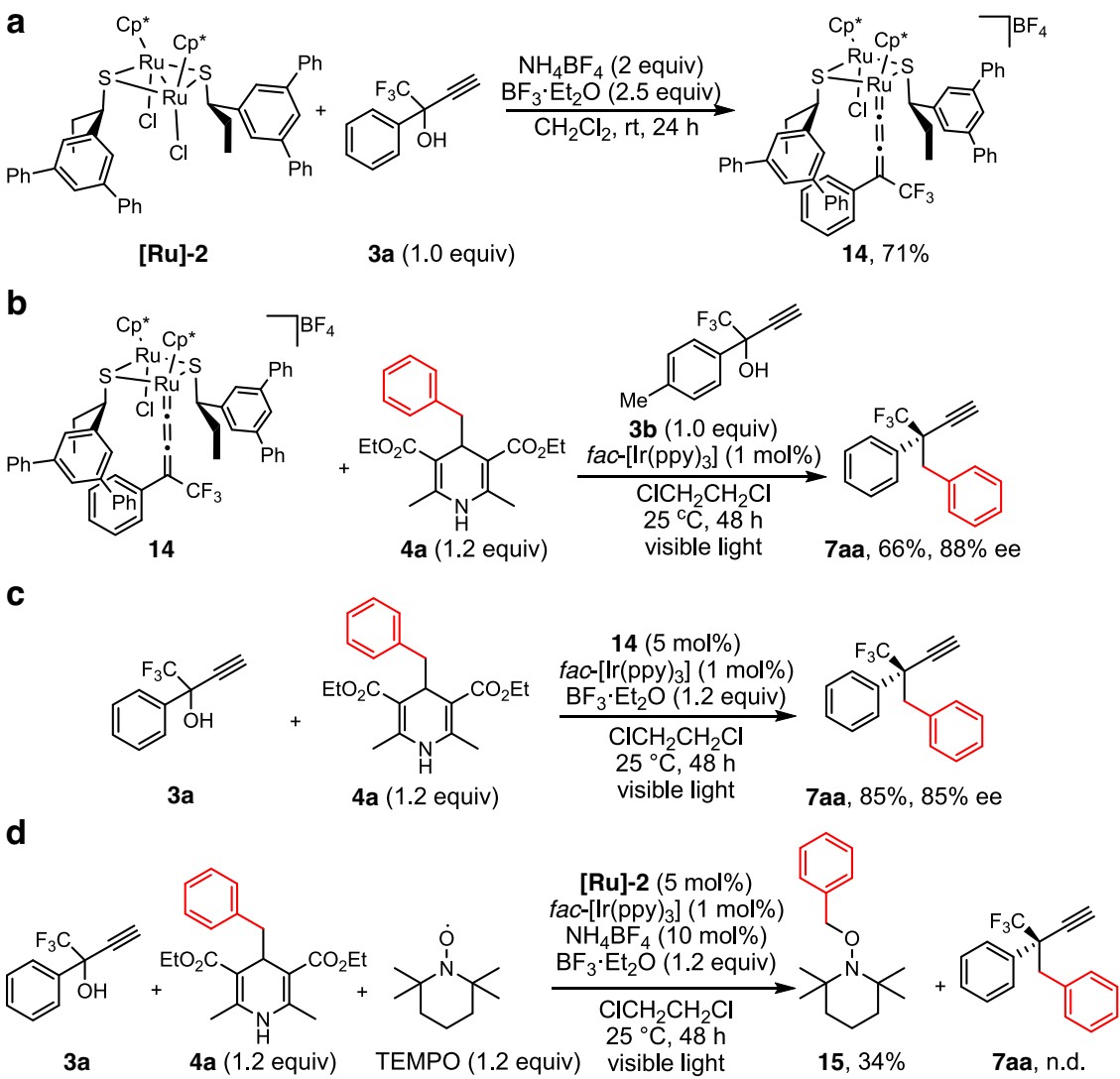

**Fig. 6 | Mechanistic studies for enantioselective propargylic alkylation.**
**a** Formation of the ruthenium–allenylidene complex in the presence of BF$_3$·Et$_2$O.
**b** Stoichiometric reaction of the ruthenium–allenylidene complex with 4-alkyl-
dihydropyridine. **c** Ruthenium–allenylidene complex-catalyzed reaction. **d** Reaction
in the presence of TEMPO.

Then, Stern–Volmer analysis for emission quenching of
*fac*-[Ir(ppy)$_3$] by **4a**, **3a**, or the ruthenium–allenylidene complex
[Cp*RuCl(μ-SMe)$_2$Ru(C=C=CHPh)Cp*]BF$_4$ (**16**), prepared by the
reaction of [{Cp*RuCl(μ-SMe)}$_2$] with **1** in the presence of NH$_4$BF$_4$
and anhydrous MgSO$_4$[55], was performed in ClCH$_2$CH$_2$Cl. As a
result, the quenching rate constants were calculated to
be $k_{4a} = (1.10 \pm 0.05) \times 10^8$ M$^{-1}$ s$^{-1}$, $k_{3a} = (5.3 \pm 0.2) \times 10^7$ M$^{-1}$ s$^{-1}$ and
$k_{16} = (1.72 \pm 0.07) \times 10^7$ M$^{-1}$ s$^{-1}$, respectively (Fig. 7a, see Supple-
mentary Information for calculations). This result revealed that
the quenching of photoexcited *fac*-[Ir(ppy)$_3$] by **4a** can occur twice
as fast as that by **3a**, and 6.5 times as fast as that by **16**.

Additional cyclic voltammetric study of *fac*-[Ir(ppy)$_3$] revealed
that the 1st oxidation wave of *fac*-[Ir(ppy)$_3$] in ClCH$_2$CH$_2$Cl appeared
at +0.25 V vs FeCp$_2^{+/0}$ as a reversible process, whereas the 1st
reduction wave appeared at −1.60 V vs FeCp$_2^{+/0}$ as an irreversible
process (Supplementary Fig. 1a). These correspond to the estimated
redox potentials of *fac*-[Ir(ppy)$_3$]*/*fac*-[Ir(ppy)$_3$]$^-$ at +0.90 V vs
FeCp$_2^{+/0}$ and *fac*-[Ir(ppy)$_3$]$^+$/*fac*-[Ir(ppy)$_3$]* at −2.25 V vs FeCp$_2^{+/0}$. On
the other hand, cyclic voltammogram of **4a** exhibited an irreversible
oxidation wave at +0.53 vs FeCp$_2^{+/0}$ (Supplementary Fig. 1b), clarify-
ing that photoexcited *fac*-[Ir(ppy)$_3$] can oxidize **4a** to afford the

radical cation **4a**$^{•+}$ via a SET process, where the C–C bond scission
further occurs to give the benzyl radical and a pyridinium cation
(Fig. 7a). As for the propargylic moiety, oxidation of **3a** was found to
be difficult to occur, but an irreversible reduction wave was observed
at −1.36 V vs FeCp$_2^{+/0}$ of **3a** in ClCH$_2$CH$_2$Cl (Supplementary Fig. 1c).
Thus, the possibility that the reduction of **3a** by the photoexcited *fac*-
[Ir(ppy)$_3$] proceeds to afford the radical anion **3a**$^{•−}$ via a SET process
leading to the formation of a propargylic radical (Fig. 7a), as has been
proposed for the formation of propargylic radicals via a SET process
of propargylic esters with an excited palladium species[37], cannot be
excluded based on the Stern–Volmer and cyclic voltammetric stu-
dies. However, radical capture experiment by TEMPO partially sup-
ports that such a SET process to generate the corresponding
propargylic radical was rather limited, but quenching of photo-
excited *fac*-[Ir(ppy)$_3$] by **3a** occurred rather via triplet state energy
transfer. Similarly, the possibility of the redox of diruthenium com-
plexes by the photoexcited *fac*-[Ir(ppy)$_3$] cannot be excluded
based on the previous CV measurements[61,62], whereas quenching
experiments have clarified that such a SET process occurs by
one order of magnitude less frequently than the SET process between
*fac*-[Ir(ppy)$_3$] and **4a**.

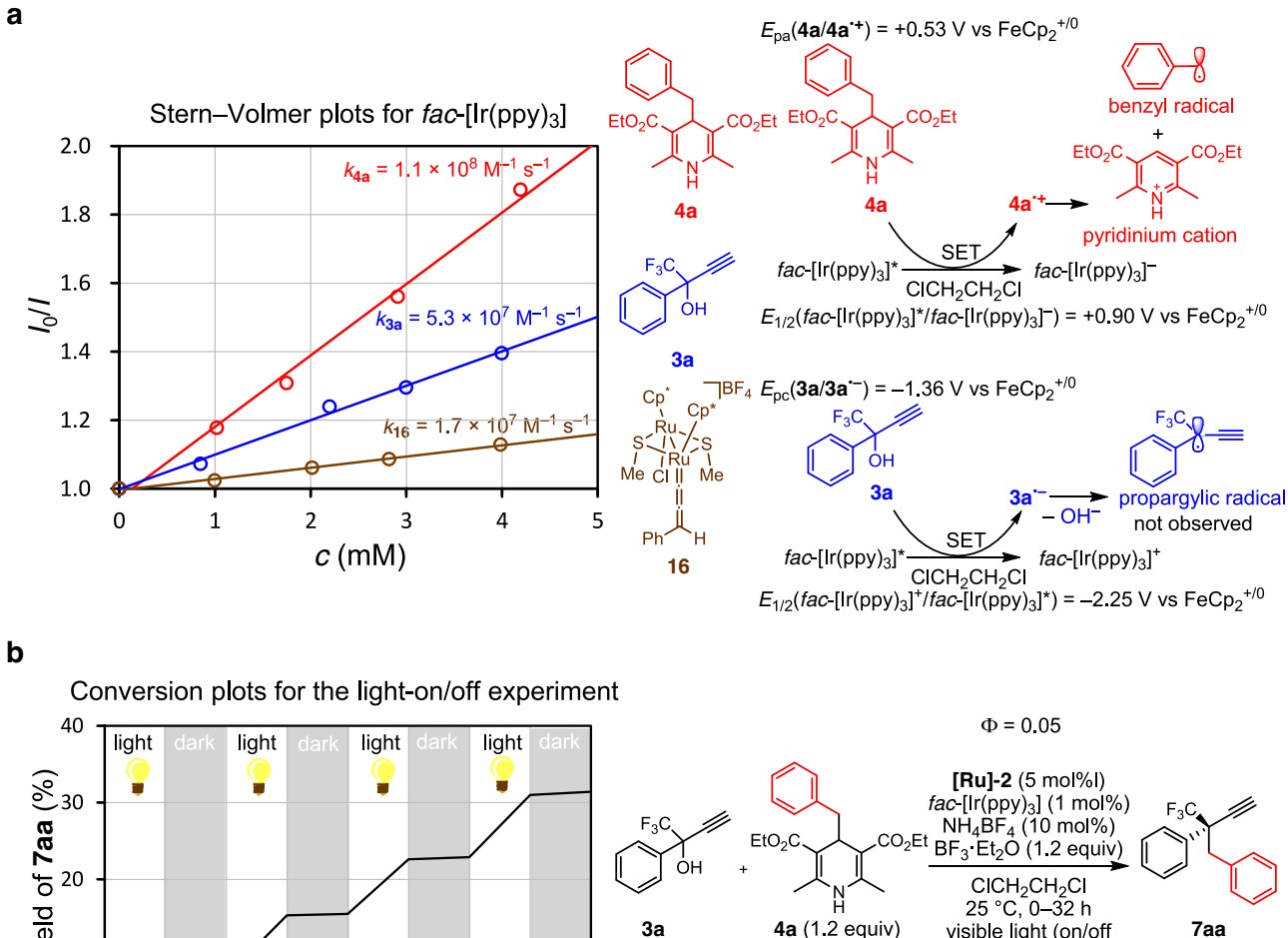

**Fig. 7 | Mechanistic studies on photoredox chemistry for enantioselective propargylic alkylation. a** Determination of quenching constants by Stern-Volmer plots and two possible quenching pathways via SET. $I_0/I$, fluorescence intensity ratio of $fac$-[Ir(ppy)$_3$]; $c$, quencher concentrations. **b** Product formation profile with light-on and -off sequences.

A light on/off experiment was further performed for propargylic alkylation of **3a** with **4a** to afford **7aa**, under the typical catalytic reaction conditions. As shown in Fig. 7b, the reaction proceeded smoothly under visible light irradiation, but did not proceed in dark, which evinced the requirement of continuous visible light irradiation is necessary, but less possibility of chain propagation participating in the main reaction pathway. In addition, time course study for propargylic alkylation of **3a** with **4a** to afford **7aa** under the typical catalytic reaction conditions was investigated (Supplementary Fig. 2), which revealed that the present dual photoredox- and ruthenium-catalyzed reaction was shown to follow the zeroth order kinetics at $k_{7aa} = (2.33 \pm 0.04) \times 10^{-7}$ M$^{-1}$ s$^{-1}$, typical for photochemical reactions under saturation of catalysts with reactants, in the optimized reaction conditions. Independently by using Hg lamp filtered at 440 nm instead of LED light, the quantum yield of alkylation of **3a** with **4a** under typical reaction conditions was determined to be $\Phi = 0.0502 \pm 0.0003$ by chemical actinometry (Supplementary Fig. 3). Both the quantum yield measurement as well as light on/off experiment have clarified that the catalytic formation of **7aa** proceeds rather via sequential redox pathways, but not via a radical chain process[63,64]. Thus, the alkyl radicals in situ are mainly formed by the SET process

between an excited-state photoredox catalyst and 4-alkyl-1,4-dihydropyridines[28,29,65].

To obtain more information on the radical reactions mediated by ruthenium−allenylidene complexes, we performed DFT calculations for the reaction of the achiral diruthenium−allenylidene complex **I** with the benzyl radical (Bn$^•$) and the pyridinium cation (PyH$^+$) (p$K_a = 2.96$ in H$_2$O)[66] generated via the scission of 4-benzyl-1,4-dihydropyridine **4a**, and the reduced iridium(II) species $fac$-[Ir(ppy)$_3$]$^−$ in ClCH$_2$CH$_2$Cl to afford a vinylidene complex, aromatized pyridine (Py), and the regenerated iridium(III) catalyst $fac$-[Ir(ppy)$_3$] (Fig. 8a; the Cartesian coordinates of the 3D structures of all species are given in the Supplementary Data 1). Here, the attack of the benzyl radical (Bn$^•$) to the $\gamma$-carbon (C$^\gamma$) atom in the monocationic diruthenium−allenylidene complex **I** gives the monocationic diruthenium−alkynyl radical complex **III** via the formation of the reactant complex **II** and the transition state **TS$_{II-III}$**. The Gibbs free energies ($\Delta G^{298K}$) of **II**, **TS$_{II-III}$**, and **III** relative to the initial state (**I** + Bn$^•$) are 4.0, 7.5 and −24.8 kcal/mol, respectively (Fig. 8a and Supplementary Fig. 5 for energy diagram and structures, respectively), indicating that these reaction steps smoothly proceed. The results of IRC calculations for **TS$_{II-III}$** are shown Supplementary Fig. 6. The change in the Mulliken spin density of

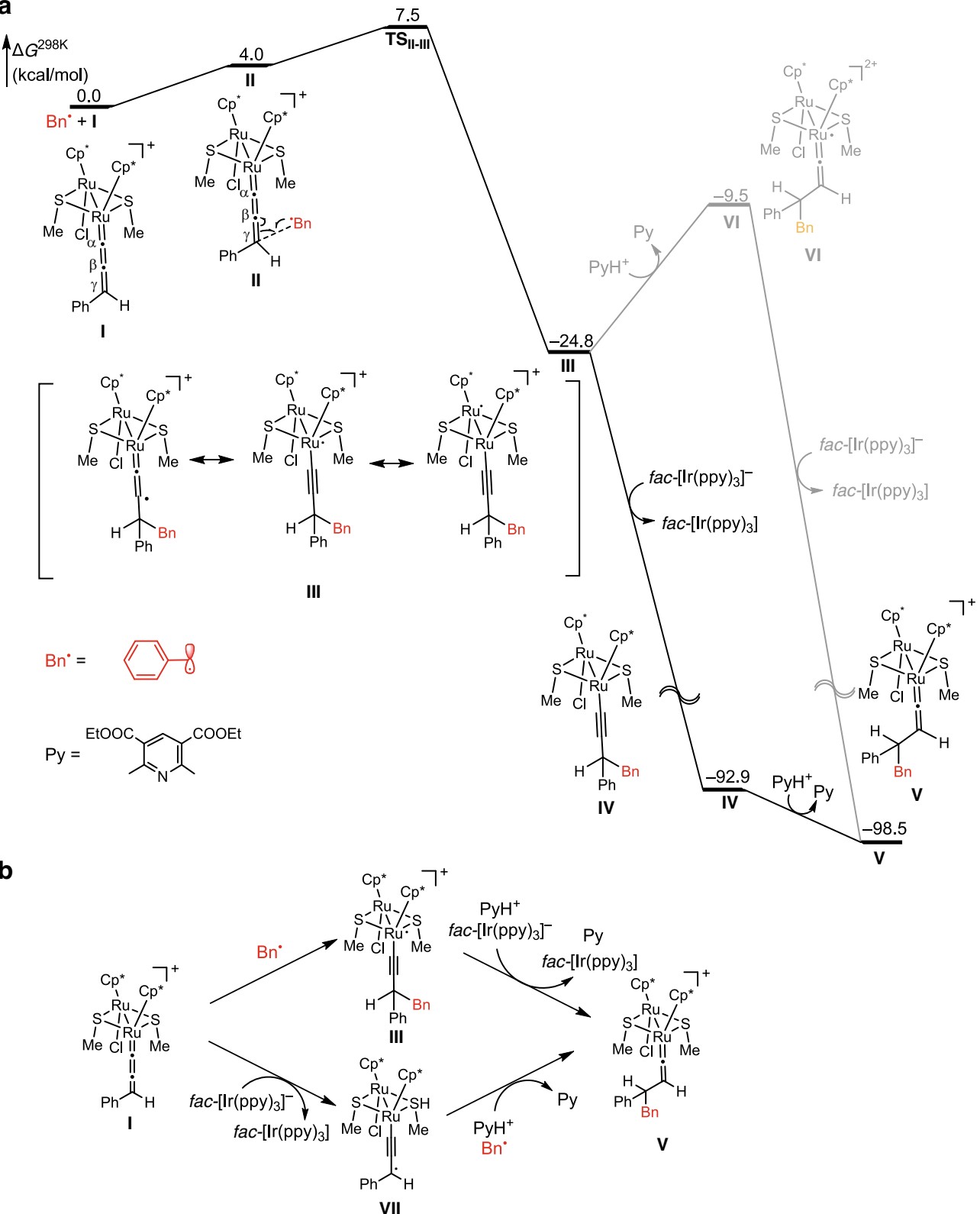

**Fig. 8 | DFT calculations for reactions of achiral methanethiolate-bridged diruthenium complexes with an alkyl radical. a** Energy diagram for the reaction of a ruthenium–allenylidene complex with benzyl radical, a reduced Ir(II) species, and a pyridinium cation. **b** Two possible reaction pathways.

atoms (Supplementary Fig. 6b) shows that the radical center moves from the carbon atom of the benzyl radical to the allenylidene region along the reaction coordinate (ruthenium and $C^\beta$ atoms). The SOMO, which has in-phase overlapping between the $C^\gamma$ atom in

the allenylidene complex and the carbon atom in benzyl radical, is gradually stabilized along the reaction coordinate, and the MO becomes doubly occupied (Supplementary Fig. 7). The spin density surface of **III** (Supplementary Fig. 8) also demonstrates

delocalization of spin density, indicating the importance of the diruthenium system.

The monocationic diruthenium–alkynyl radical complex III is then exergonically reduced by $^2$[Ir(ppy)$_3$]$^-$ to form the neutral diruthenium–alkynyl complex IV (III + $fac$-[Ir(ppy)$_3$]$^-$ ($\Delta G^{298K} = -24.8$ kcal/mol) → IV + $fac$-[Ir(ppy)$_3$] ($\Delta G^{298K} = -92.9$ kcal/mol)), while III is endergonically protonated by protonated pyridine to afford the dicationic diruthenium–vinylidene complex VI (III + PyH ($\Delta G^{298K} = -24.8$ kcal/mol) → VI + Py ($\Delta G^{298K} = -9.5$ kcal/mol)). Thus, it is more likely that III is first reduced to afford IV, and then IV is protonated to afford the diruthenium–vinylidene complex V ($\Delta G^{298K} = -98.5$ kcal/mol). The protonation of IV was calculated to be exergonic by 5.6 kcal/mol. Our present DFT calculations indicate that the attack of benzyl radical to the C$^\gamma$ atom in the ruthenium–allenylidene complex followed by the reduction and protonation easily gives the ruthenium–vinylidene complex.

It should be carefully paid attention that two reaction pathways can be possible for the formation of the vinylidene complex V from the allenylidene complex I: one is the attack of Bn· to the allenylidene complex I to form the monocationic alkynyl radical complex III at first, followed by the one-electron reduction by $fac$-[Ir(ppy)$_3$]$^-$ to generate the vinylidene complex V (upper pathway in Fig. 8b) as has been shown by the DFT calculations in Fig. 8a; the other is one-electron reduction of the allenylidene complex I by $fac$-[Ir(ppy)$_3$]$^-$ to form the neutral alkynyl radical complex VII at first, followed by the attack of Bn· (lower pathway in Fig. 8b). Indeed, mononuclear ruthenium allenylidene complexes were known to be reduced by strong reducing reagents such as cobaltocene to afford the radical species[67], which were known to react with radical reagents[68,69]. However, our group previously found that the thiolate-bridged diruthenium complex [{Cp*RuCl(μ-SMe)}$_2$] underwent reductive coupling of propargylic alcohols including 1 to afford 1,5-hexadiynes under reductive conditions[44]. Here, a neutral alkynyl radical complex VII was generated in situ by the reaction of I with a trace amount of oxygen, which reacted with another radical species in the presence of a proton source to afford the corresponding propargylic homo-coupled product (Fig. 8b). However, such compounds were not obtained at all, both for the reaction of 3a with 4a (Table 1, entry 8) or for the reaction of 1a with 4a in the present reaction conditions except for using [{Cp*RuCl(μ-SMe)}$_2$] as a ruthenium catalyst instead of [Ru]-1 or [Ru]-2. The corresponding propargylic radicals were also not trapped by TEMPO (Fig. 8b), which may support the reaction pathway via the attack of benzyl radical at first, followed by the one-electron reduction. However, we cannot exclude the possibility of the other reaction pathway only by the TEMPO experimental result.

Based on the mechanistic studies, a plausible catalytic reaction pathway can be drawn as depicted in Fig. 9a, containing two catalytic cycles: the photoredox catalytic cycle and the ruthenium catalytic cycle. In the photoredox cycle, the iridium catalyst $fac$-[Ir(ppy)$_3$] ([Ir]) is excited under visible light irradiation to afford a photoexcited iridium catalyst ([Ir]*). Then, a SET process between [Ir]* and 4-alkyl-1,4-dihydropyridine (4) occurs to give the reduced iridium catalyst ([Ir]$^-$), an alkyl radical (R·), and a pyridinium cation (PyH$^+$) via C−C bond scission. On the other hand, in the ruthenium cycle, coordination of the propargylic alcohol (3) to the coordinatively unsaturated species (G), obtained by the reaction of [Ru]-1 or [Ru]-2 with NH$_4$BF$_4$, occurs first to afford the π-alkyne complex (H), leading to the formation of the vinylidene complex (I) via 1,2-proton transfer. Then, dehydration accelerated by BF$_3$·Et$_2$O occurs to give the allenylidene complex (J). Then, alkyl radical attacks at the γ-position of the allenylidene ligand of J from $Re$ face to afford the alkynyl radical complex (K). The alkynyl radical complex K, which is stabilized by the diruthenium core acting as an electron pool to furnish the radical redox reaction, is then reduced by the iridium(II) species [Ir]$^-$ through single electron transfer to afford the alkynyl complex (L). Then protonation at the β-position

occurs, presumably by the pyridinium cation PyH$^+$ to afford the vinylidene complex (M), followed by the proton transfer to afford the π-alkyne complex (N). Finally, ligand exchange with propargylic alcohol 3 gives the product (7) and regenerates the starting π-alkyne complex H via the formation of coordinatively unsaturated complex G.

As for the asymmetric induction, the enantioselectivity of the products presented in this manuscript is the same as that observed for the enantioselective propargylic phosphinylation of propargylic alcohols bearing both aryl and CF$_3$ moieties at the propargylic position with phosphorus-centered nucleophiles in the presence of [Ru]-1 or [Ru]-2 as a catalyst[13]. The mechanism of asymmetric induction has been recently elucidated in details by DFT calculations for the enantioselective nucleophilic propargylic substitution reaction of 3a with a nucleophile in the presence of [Ru]-2 as a catalyst, which has revealed the existence of intramolecular π/π and C−H/F interactions among the allenylidene, the optically active thiolate, and pentamethylcyclopentadienyl ligands in the diruthenium–allenylidene complex[34]. Accordingly, alkyl radicals can almost only approach from the $Re$ face of the diruthenium–allenylidene intermediate (from backward in Fig. 9b), similarly to that proposed for the enantioselective propargylic substitution reactions of derived propargylic alcohols bearing both aryl and CF$_3$ moieties with nucleophiles[34], leading to the formation of (R)-isomer as the major product.

## Discussion

We have developed the photoredox- and diruthenium-catalyzed enantioselective propargylic alkylation of propargylic alcohols with 4-alkyl-1,4-dihydropyridines under ambient conditions and visible light irradiation. Especially, we have succeeded in the construction of a quaternary stereogenic carbon center at the propargylic position. This is the first successful example of transition metal-catalyzed enantioselective propargylic substitution reactions with free radicals, demonstrating that free radicals generated by the photoredox catalysis can act as formal nucleophilic reagents to afford substituted products that cannot be achieved by reactions with ionic nucleophiles. The key reaction step is the attack of an alkyl radical to the γ-carbon of the ruthenium-allenylidene complex bearing optically active thiolate ligands. The dual photoredox- and diruthenium-catalytic system has also controlled the radical reactions with the photoredox catalyst generating free radicals and the diruthenium catalyst capturing free radicals, keeping the concentration of free radicals constantly low. The usage of 4-alkyl-1,4-dihydropyridine skeletons as milder alkylation reagents has successfully expanded toward the enantioselective propargylic substitution reactions, adding an entry for enantioselective radical processes[1,2]. We believe this work provides another fundamental method for organic syntheses under mild reaction conditions.

## Methods

### General procedure for the preparation of chiral propargylic alkylation products

In an oven dried 20 mL Schlenk flask were placed [Ru]-2 (6.3 mg, 0.0050 mmol) and NH$_4$BF$_4$ (1.1 mg, 0.010 mmol) under N$_2$. Anhydrous ClCH$_2$CH$_2$Cl (2.0 mL) was added, and then the mixture was magnetically stirred at room temperature for 30 min. Then, 1,1,1-trifluoro-2-phenylbut-3-yn-2-ol (3a) (20.0 mg. 0.10 mmol), diethyl 4-benzyl-2,6-dimethyl-1,4-dihydropyridine-3,5-dicarboxylate (4a) (41.2 mg, 0.12 mmol), $fac$-[Ir(ppy)$_3$] (0.7 mg, 0.0011 mmol), and BF$_3$·Et$_2$O (15 μL, 17 mg, 0.12 mmol) were added under N$_2$ at room temperature. The reaction flask was placed in an As One LTB-125 constant low temperature water bath set at 25 °C, and was illuminated from the bottom of the bath with an Aitech System TMN100 × 120−22WD 12 W white LED lamp (400−750 nm) at a distance of ~2 cm from the light source for 48 h. The volatiles were

**Fig. 9 | Diruthenium-catalyzed enantioselective alkylation of propargylic alcohols with alkyl radicals. a** Plausible catalytic reaction pathways. **b** Asymmetric induction via intramolecular π/π and C−H/F interactions within an allenylidene intermediate.

removed in vacuo, and the residue was purified by column chromatography (SiO₂) with *n*-hexane as an eluent to afford (*R*)-(2-(trifluoromethyl)but-3-yne-1,2-diyl)dibenzene ((*R*)-**7aa**) as a colorless oil (22.2 mg, 0.081 mmol, 81% yield based on the amount of **3a**). The enantiomeric excess of **7aa** was determined by HPLC analysis; DAICEL Chiralpak OJ-H, hexane/$^i$PrOH = 99/1, flow rate = 0.5 mL/min, $\lambda$ = 220 nm, retention time: 14.6 min (major) and 19.9 min (minor), 94% ee.

## Data availability

The crystallographic data for the structure generated in this study have been deposited at the Cambridge Crystallographic Data Centre under deposition number CCDC 2172478 ((*R*)-**7lr**). Copies of the data can be obtained free of charge from www.ccdc.cam.ac.uk/structures/.

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

## Acknowledgements

We acknowledge Grants-in-Aid for Scientific Research (Grants 20H05671, 20K05680, 20K21203, and 22K19041) from JSPS and MEXT. Y.Z. is a recipient of the JSPS Research Fellowships for Young Scientists.

## Author contributions

Y.Z., Y.T., and Y.N. designed and analyzed the experiments. Y.Z. ran all experiments. Y.T. conducted X-ray crystallographic and CV measurements. K.S. conducted DFT calculations. Y.Z., Y.T., K.S., and Y.N. wrote the manuscript. Y.Z., Y.T., S.K., K.S., and Y.N. contributed to discussions.

## Competing interests

The authors declare no competing interests.
