## [Peer Review File · Nature Communications]

REVIEWER COMMENTS

Reviewer #1 (Remarks to the Author):

This manuscript presents an enantioselective propargylic substitution reaction catalyzed a diruthenium complex. Although the substrate scope is a bit limited, the reaction is interesting. Mechanistic investigations were also well performed. I support the publication of this manuscript. My comments are the following.

1. The transition state model J in Fig. 6b is not consistent with the geometry of TSII-III in Fig. 5a. The allenylidene plane in TSII-III is perpendicular to what was drawn in J and the Re face is not from backward but right side. A more reasonable model should be provided.
2. The CF₃ group plays a critical role in enantioselectivity in this work and some of the previous works from the authors' group. The substrate with -CF₂H, instead of -CF₃, gave lower ee. What other groups was tested? What really matters? Electronic effect or steric effect? If only CF₃ works, the substrate scope is limited.
3. The units, such as equiv., mol%, and mmol, in figures 3 and 4 are not used in a consistent way.
4. In the product of pathway ii in Fig. 2, R1 and R2 should be Ph and H.

Reviewer #2 (Remarks to the Author):

This manuscript by Tanabe, Sakata, and Nishibayashi described a ruthenium catalyzed enantioselective propargylic substitution reactions with visible light-generated alkyl radicals to generate propargylic alkylated products containing a quaternary stereogenic carbon center at the propargylic position with a high enantioselectivity. The control experiments proved the rationality of the mechanism speculation. While several examples of ruthenium-catalyzed enantioselective propargylic reactions have been reported by the authors. So, I think the catalytic system and the concept are not high novelty. Meanwhile, the scope of the reaction was based primarily on the benzylic radical species, and I feel the substrate scope is limited. So, if the author could extend the substrate scope with other radical species, I could recommend acceptance this work.

Reviewer #3 (Remarks to the Author):

The manuscript describes catalytic radical propargyl substitution reactions between propargylic alcohols and radicals formed under catalytic photoredox conditions from alkyl-1,4-dihydropyridines. The method employs a dual iridium-based photoredox / ruthenium catalyst system, where the photocatalyst generates alkyl radicals to be caught by the ruthenium catalyst, which also activates the propargylic alcohol substrates. When a propargylic alcohol and an alkyl-1,4-dihydropyridine (1.2 equivalent) was reacted for 48 h at room temperature in presence of NH_4BF_4 (10 mol%), BF_3 (1.2 equivalents), a chiral thiolate-bridged ruthenium catalyst (5 mol%) and $\text{fac-}[\text{Ir}(\text{ppy})_3]$ (1 mol%) in $\text{ClCH}_2\text{CH}_2\text{Cl}$, the corresponding substitution products were isolated in 83-65% isolated yields and enantiomeric excesses ranging from 98-83%. Experiments and calculations were performed to establish a mechanism for the title reaction.

In general, the reaction is interesting, because it provides stereoselective access to quaternary carbon stereocenters, which is a challenging reaction intermolecularly. The reaction is, in general, known for achiral or racemic syntheses (ref #71 of the manuscript) or related protocols (ref #56). However, the manuscript advances the knowledge in the area with respect to the stereochemistry. As such, the manuscript is publishable, however, the authors should address the following points.

In Figure 4a, a stoichiometric experiment between a Ru allenylidene complex, the dihydropyridine and a propargylic alcohol different from the one used in the Ru allenylidene, but without BF_3 is described. The extra added propargylic alcohol did not show up in the product, and the authors concluded that the BF_3 is required for allenylidene formation from the propargylic alcohol. First, it could be that the preformed allenylidene complex just reacts faster with the radicals than the formation of a new allenylidene complex with the extra added propargylic alcohol would take. Furthermore, as shown in Table 1, entry 1, the reaction works without BF_3 for a secondary propargylic alcohol. The mechanism may be different here, however, entry 23 in Table 1 would be much better evidence that BF_3 is essential. If BF_3 assists with the formation in the allenylidene intermediate or has another role can in my opinion not unambiguously be determined based on the experiment. A better way to examine the role of BF_3 may be whether it is required to synthesize the allenylidene complex 14.

In Figure 4b, a catalysis experiment with a catalytic amount of an allenylidene complex is described, and the authors write, that this is evidence that the allenylidene must be a key intermediate. However, it cannot be excluded that the allenylidene catalyst is actually a precatalyst. Obviously, complex 14 is stable. However, stable compounds are not necessarily good intermediates in catalytic cycles, they would slow down or can even completely inhibit a cycle.

In Figure 4c, the title reaction is performed in the presence of the radical scavenger TEMPO. Only benzylated TEMPO was observed in the reaction mixture, and the authors conclude that only benzyl radicals form and no propargyl radicals. However, it may just be that TEMPO much more efficiently captures the benzyl radical compared to the tertiary, more sterically congested propargyl radical. References 102 and 103 can only limitedly serve for comparison, as in these cases, secondary propargylic halides (not alcohols) are employed and in these references, there are no substrates that even could form a benzyl radical. As such, the findings of the authors are not really “in contrast” to what references 102 and 103 suggest (I cannot see propargylic esters in references 102 and 103, as written by the authors). I think under the reaction conditions the formation of benzyl radicals is not unlikely, but the formation of propargylic radicals cannot be entirely excluded. Does TEMPO inhibit the reaction?

Likewise, the mechanistic experiment employing deuterated dihydropyridine (Figure 4d) is not very meaningful. In the product, 30% of the acetylenic protons are deuterated, and the authors conclude that mechanistically, the proton for the protonation of the potential alkynyl intermediate must originate from the deuterated dihydropyridine. This is a low deuterium content to derive mechanistic proposals. Moreover, as pointed out by the authors, isotope scrambling could happen either on the acetylenic position, but also beforehand between the deuterated pyridine and NH_4BF_4 or the propargylic alcohol substrate or the water formed during the reaction. The acetylenic proton can, in general, come from the water, NH_4BF_4 , the propargylic alcohol and the solvent or the dihydropyridine, or, most likely, form a combination of those. As such, I think the deuterium-labeling experiment contributes only little to the mechanistic picture.

It seems the calculations in Figure 5 were performed with secondary propargylic alcohols without a CF_3 group. Could it be that the mechanism is different when a tertiary, CF_3 substituted propargylic alcohol (the actual substrate) is employed? On the very bottom right, a “TEMPO trapped” substrate is labeled as “not observed”, however, the authors would have observed the product originating from the tertiary propargylic alcohol actually employed in the reaction. I am not sure if the two pathways in 5b can be distinguished by a TEMPO experiment. The lack of formation of the homo-coupled product may not exclude the pathway in Scheme 5b, because it may just be more difficult to homodimerize two tertiary carbon centers. In reference 87, secondary propargylic alcohols are employed, and a comparison with the mechanism in the current manuscript is, thus, shaky.

Figure 1a (i) and (ii), “allylic positions” misspelled. Page 14, “syntetic” and “congainng” misspelled. Page 15, “allenylinde” misspelled. Page 8, “propagylic” misspelled. Page 22, “leadgin” misspelled. In some references (like #12) the phrase “et al.” is used, however, I think all authors should be listed.

The followings are our answers to the comments.

As for the comments by Reviewer 1

- (1) Reviewer 1 pointed out "The transition state model **J** in Fig. 6b is not consistent with the geometry of TS_{II-III} in Fig. 5a. The allenylidene plane in TS_{II-III} is perpendicular to what was drawn in **J** and the *Re* face is not from backward but right side. A more reasonable model should be provided" As pointed by Reviewer 1, the structures of the thiolate and allenylidene ligands are different between the transition state model in **J** in Fig. 9b of the revised manuscript (Fig. 6b of the original manuscript) and the geometry of TS_{II-III} in Fig. 8a of the revised manuscript (Fig. 5a of the original manuscript). The thiolate and allenylidene ligands of the transition state in **J** in Fig. 9b of the revised manuscript (Fig. 6b of the original manuscript) are the chiral one ((*R*)-SCH(Et)C₆H₃Ph₂-3,5) and that derived from a CF₃-containing propargylic alcohol that induces enantioselectivity, whereas the thiolate and allenylidene ligand in Fig. 8a of the revised manuscript (Fig. 5a of the original manuscript) are the achiral methanethiolate (SMe) and that derived from a secondary propargylic alcohol without the CF₃ moiety. As to the structure of **J** shown in Fig. 9b of the revised manuscript (Fig. 6b of the original manuscript), we previously reported the DFT calculations of the attack of nucleophiles from the *Re* face of the allenylidene ligand in **J** (ref. 34 of the revised manuscript (ref. 66 of the original manuscript): *ACS Omega* **41**, 36634 (2022)). On the other hand, the DFT calculations in this present study as shown in Fig. 8a of the revised manuscript (Fig. 5a of the original manuscript) were carried out for the reaction with an alkyl radical, thus various methanethiolate-bridged diruthenium complexes as reactive intermediates were used for the discussions on the reaction pathway. In order to make it clear that asymmetric induction is not discussed, but the reaction of achiral methane-thiolate-bridged diruthenium complexes with an alkyl radical is discussed in the present manuscript, we have newly modified the figure title of Fig. 8a of the revised manuscript (Fig. 5a of the original manuscript) as "DFT calculations for reactions of achiral methanethiolate-bridged diruthenium complexes with an alkyl radical." Thank you very much for your suggestions.
- (2) Reviewer 1 pointed out "The CF₃ group plays a critical role in enantioselectivity in this work and some of the previous works from the authors' group. The substrate with -CF₂H, instead of -CF₃, gave lower ee. What other groups was tested? What really matters? Electronic effect of steric effect? If only CF₃ works, the substrate scope is limited." As discussed in the first paragraph, page 5, Table 1, entries 1–6, and Fig. 2a of the revised manuscript (page 6 and Table 1, entries 1–6 of the original manuscript), we already carried out reactions with -CH₃ and -H instead of -CF₃, giving the corresponding product with a low ee (48% ee) and almost no corresponding desired product, respectively. In addition, we already carried out the reaction with -CF₂Cl instead of -CF₃ to obtain the corresponding product only in a low yield. However unfortunately, we could not determine ee of the product. As shown in the substrate scope in the present manuscript, we believe that the use of CF₃-containing propargylic alcohols is necessary to achieve high enantioselectivity in the propargylic alkylation with alkyl radicals. In fact, a similar tendency was observed in our previous enantioselective propargylic phosphinylation with phosphine oxides (ref. 13 of the revised manuscript (ref. 21 of the original manuscript): *Angew. Chem. Int. Ed.* **60**, 11231 (2021)). As pointed

out by Reviewer 1, we achieved the high enantioselectivity when we used CF₃-containing propargylic alcohols as substrates. However, we would like to emphasize that the content of the present manuscript provides a great advance in the enantioselective propargylic substitution reactions. Several successful examples of enantioselective propargylic substitution reactions with various nucleophiles have been developed by our and other research groups in the world, however, the result described in the present manuscript is the last remaining piece of enantioselective propargylic substitution reactions with free alkyl radicals. Thank you very much for your careful reading.

- (3) Reviewer 1 pointed "The units, such as equiv., mol%, and mmol, in figures 3 and 4 are not used in a consistent way." According to the suggestion, we have corrected all of the original descriptions in Figs. 3, 4, 6, and 7 of the revised manuscript (Fig. 3 and 4 of the original manuscript). Thank you very much for your careful reading.
- (4) Reviewer 1 pointed out "In the product of pathway ii in Fig. 2, R1 and R2 should be Ph and H." According to the suggestion, we have corrected all of the original descriptions in Fig. 2 of the revised manuscript. Thank you very much for your careful reading.

As for the comments by Reviewer 2

- (1) Reviewer 2 pointed out "While several examples of ruthenium-catalyzed enantioselective propargylic reactions have been reported by the authors. So, I think the catalytic system and the concept are not high novelty." As pointed out by Reviewer 2, we and other research groups reported several successful examples of enantioselective propargylic substitution reactions with various nucleophiles. However, we would like to emphasize that the content of the present manuscript provides a great advance in the enantioselective propargylic substitution reactions because the result described in the present manuscript is the last remaining piece of enantioselective propargylic substitution reactions with free alkyl radicals. In order to clarify such novelty, we have newly added the following explanation to the Abstract of the revised manuscript as "The result described in this paper provides the first successful example of transition metal-catalyzed enantioselective propargylic substitution reactions with free alkyl radicals." Thank you very much for your careful reading.
- (2) Reviewer 2 pointed out "Meanwhile, the scope of the reaction was based primarily on the benzylic radical species, and I feel the substrate scope is limited." As shown in Fig. 3a of the revised manuscript (Table 2a of the original manuscript), the substrate scope is primarily based on the benzylic radical species. However, as discussed in the last paragraph, page 9 of the revised manuscript (the second paragraph, page 14 of the original manuscript), we already investigated the reactions with 1,4-dihydropyridine derivatives bearing not only pyridine-, ether-, and amine-containing methyl groups but also secondary alkyl and long chain primary alkyl groups as shown in Fig. 3b of the revised manuscript. We observed only trace amounts of the desired products in all cases. This result indicates that only benzyl radical species are available as radical sources for the enantioselective propargylic alkylation with simple alkyl radicals. In order to clarify such scope limitations, we have revised a part of Table 2 of the original manuscript as Fig. 3b of the revised

manuscript. Namely, we have newly added unsuccessful dihydropyridines as sources of alkyl radical species in Fig 3b. As pointed out by Reviewer 2, we achieved the high enantioselectivity when we used benzyl radical species as alkylation reagents. However, we would like to emphasize that the content of the present manuscript provides a great advance in the enantioselective propargylic substitution reactions. Several successful examples of enantioselective propargylic substitution reactions with various nucleophiles have been developed by our and other research groups in the world, however, the result described in the present manuscript is the last remaining piece of enantioselective propargylic substitution reactions with free alkyl radicals. Thank you very much for your careful reading.

As for the comments by Reviewer 3

- (1) Reviewer 3 pointed out "In Figure 4a, a stoichiometric experiment between a Ru allenylidene complex, the dihydropyridine and a propargylic alcohol different from the one used in the Ru allenylidene, but without BF₃ is described. The extra added propargylic alcohol did not show up in the product, and the authors concluded that the BF₃ is required for allenylidene formation from the propargylic alcohol. First, it could be that the preformed allenylidene complex just reacts faster with the radicals than the formation of a new allenylidene complex with the extra added propargylic alcohol would take. Furthermore, as shown in Table 1, entry 1, the reaction works without BF₃ for a secondary propargylic alcohol. The mechanism may be different here, however, entry 23 in Table 1 would be much better evidence that BF₃ is essential. If BF₃ assists with the formation in the allenylidene intermediate or has another role can in my opinion not unambiguously be determined based on the experiment. A better way to examine the role of BF₃ may be whether it is required to synthesize the allenylidene complex **14**." According to the suggestions by Reviewer 3, we have newly carried out the reaction of **[Ru]-2** with the CF₃-containing propargylic alcohol **3a** in the presence of NH₄BF₄ and BF₃·Et₂O, in place of anhydrous MgSO₄, to afford the corresponding allenylidene complex **14** in 71% yield. This result indicates that BF₃ worked as the dehydration reagent to afford the corresponding allenylidene complexes from reactions with propargylic alcohols containing a CF₃ group at the propargylic position. However, as pointed out by Reviewer 3, we cannot completely exclude other possible roles of BF₃ in the present reaction system. Based on these experimental results and comments pointed out by Reviewer 3, we have deleted the following descriptions in the first paragraph, page 2 of the original manuscript: "Here, the corresponding propargylic alkylated product **7ba**, supposed to be produced by the reaction of **3b** with **4a**, was not obtained at all. Namely, **3b** only worked to promote the liberation of **7aa** from the ruthenium complex by ligand exchange, but further transformation did not take place for **3b** coordinated to the ruthenium catalyst **[Ru]-1**, when BF₃·Et₂O was not present. These results clearly indicate that BF₃·Et₂O does not participate in the alkylation of the allenylidene complex, but is necessary for the formation of allenylidene complex by accelerating the dehydration process from propargylic alcohols." in the revised manuscript. Instead, we have newly added the formation of the allenylidene complex **14** from the reaction of **[Ru]-2** with the CF₃-containing propargylic alcohol **3a** in the presence of BF₃·Et₂O as Fig. 6a of the revised manuscript. We have

also newly added some explanations to the second paragraph, page 12 of the revised manuscript as follows: "These results indicate that $\text{BF}_3 \cdot \text{Et}_2\text{O}$ is not concerned with the alkylation of the allenylidene complex but likely with the dehydration process for the formation of allenylidene complex at least for propargylic alcohols containing CF_3 group at the propargylic position." On the other hand, Reviewer 3 is wondering why the corresponding allenylidene complex derived from the additional propargylic alcohol **3b** was not obtained in the stoichiometric reaction of the allenylidene complex **14** with the dihydropyridine **4a** under photoredox reaction conditions. In this stoichiometric reaction, benzyl radical generated from **4a** under photoredox reaction conditions first attacks the γ position of the allenylidene complex of **14** to afford the corresponding vinylidene complex. Then, the vinylidene complex is converted to the π -alkyne complex via a proton transfer. Finally the ligand exchange of the coordinated alkyne with the additional propargylic alcohol **3b** occurs to give the corresponding propargylic alkylated product **7aa**. The reason why we added the additional propargylic alcohol **3b** to the stoichiometric reaction of the allenylidene complex **14** with the dihydropyridine **4a** is the promotion of the ligand exchange of the coordinated alkyne to give the corresponding propargylic alkylated product **7aa**. After the stoichiometric reaction, the corresponding allenylidene and vinylidene complexes derived from the additional propargylic alcohol **3b** may be formed in the reaction mixture. As a result, we did not observe the formation of propargylic alkylated product derived from the additional propargylic alcohol **3b** in the stoichiometric reaction of the allenylidene complex **14** with the dihydropyridine **4a** under photoredox reaction conditions as shown in Fig. 6b of the revised manuscript (Fig. 4a of the original manuscript). Thank you for your valuable suggestion.

- (2) Reviewer 3 pointed out "In Figure 4b, a catalysis experiment with a catalytic amount of an allenylidene complex is described, and the authors write, that this is evidence that the allenylidene must be a key intermediate. However, it cannot be excluded that the allenylidene catalyst is actually a precatalyst. Obviously, complex **14** is stable. However, stable compounds are not necessarily good intermediates in catalytic cycles, they would slow down or can even completely inhibit a cycle." As to the formation of allenylidene complex, we must note that the addition of dehydration reagents is always necessary to isolate the corresponding allenylidene complex as the stable intermediate as shown in Fig. 6a of the revised manuscript, because there is an equilibrium between the vinylidene complex derived from **[Ru]-2** and **3a** and the allenylidene complex **14** and water. Based on the equilibrium between the vinylidene and allenylidene complexes, the allenylidene complex does not exist as a stable reactive intermediate under the catalytic reaction conditions. In the catalytic reaction, the allenylidene complex smoothly reacts with the benzyl radical to afford the propargylic alkylated product via the formation of vinylidene complex. Thus, the allenylidene complex is only stable in the stoichiometric reaction, but not stable in the catalytic reaction. We explained these points in our previous papers (typically ref. 31 of the revised manuscript (ref. 63 of the original manuscript): *J. Am. Chem. Soc.* **127**, 9428 (2005) and ref. 56 of the revised manuscript: *Chem. Eur. J.* **11**, 1433 (2005)). Thank you for your valuable suggestion.
- (3) Reviewer 3 pointed out "In Figure 4c, the title reaction is performed in the presence of the radical scavenger TEMPO. Only benzylated TEMPO was observed in the reaction mixture, and the authors conclude that only benzyl radicals form and no propargyl radicals. However, it may just be that TEMPO much more efficiently captures the benzyl radical compared to

the tertiary, more sterically congested propargyl radical. References 102 and 103 can only limitedly serve for comparison, as in these cases, secondary propargylic halides (not alcohols) are employed and in these references, there are no substrates that even could form a benzyl radical. As such, the findings of the authors are not really "in contrast" to what references 102 and 103 suggest (I cannot see propargylic esters in references 102 and 103, as written by the authors). I think under the reaction conditions the formation of benzyl radicals is not unlikely, but the formation of propargylic radicals cannot be entirely excluded. Does TEMPO inhibit the reaction?" First of all, we are very sorry for the wrong citation of references 102 and 103 in the original manuscript. We would like to change references 102 and 103 of the original manuscript to ref. 37 of the revised manuscript (ref. 71 of the original manuscript: *Chem. Commun.* **56**, 12957 (2020)). In this reference, photoredox/palladium co-catalyzed propargylic benzylation with secondary propargylic carbonates bearing an internal alkyne moiety is reported, where the formation of a TEMPO-trapped propargylic alcohol was confirmed by GC-MS, although dimerization of the benzyl radical was also detected by GC-MS. As pointed out by Reviewer 3, we cannot exclude the formation and participation of propargylic radicals in our reaction system. It is also likely that TEMPO more efficiently captures the benzyl radical compared to the tertiary, more sterically congested propargylic radicals under our reaction system. Taking the suggestions by Reviewer 3 into consideration, we have deleted the descriptions in the first paragraph, page 21 of the original manuscript: "In contrast, formation of other TEMPO-trapped compounds was not observed, demonstrating that the formation of benzyl radicals preferentially occurred under photoredox catalytic conditions, whereas formation of propargylic radicals did not take place. This experimental result is in contrast to that obtained for photoredox- and palladium-catalyzed alkylation of a propargylic ester with **4a**, where TEMPO did not trap a benzyl radical but trapped a propargylic radical, which was proposed to be generated via a SET process between an excited palladium species and a propargylic ester in the presence of a base under visible light irradiation." in the revised manuscript. Instead, we have newly added some explanations to the second paragraph, page 13 of the revised manuscript as follows: "Formation of other TEMPO-trapped compounds was not observed from the reaction mixture. At present, we cannot exclude the possibility of the formation of propargylic radicals as reactive intermediates because we used tertiary propargylic alcohols bearing the CF₃ moiety as the substrate, where the corresponding propargylic radicals are difficult to be trapped by TEMPO." Thank you for your valuable suggestion.

- (4) Reviewer 3 pointed out "Likewise, the mechanistic experiment employing deuterated dihydropyridine (Figure 4d) is not very meaningful. In the product, 30% of the acetylenic protons are deuterated, and the authors conclude that mechanistically, the proton for the protonation of the potential alkynyl intermediate must originate from the deuterated dihydropyridine. This is a low deuterium content to derive mechanistic proposals. Moreover, as pointed out by the authors, isotope scrambling could happen either on the acetylenic position, but also beforehand between the deuterated pyridine and NH₄⁺BF₄⁻ or the propargylic alcohol substrate or the water formed during the reaction. The acetylenic proton can, in general, come from the water, NH₄BF₄, the propargylic alcohol and the solvent or the dihydropyridine, or, most likely, form a combination of those. As such, I think the deuterium-labeling experiment contributes only little to the mechanistic picture." Taking the

comments by Reviewer 3 into consideration, we have deleted the original descriptions in the second paragraph, page 21 of the original manuscript: “Next, the propargylic alkylation of **3a** with 1-deuterated 4-benzyl-1,4-dihydropyridine (**4a-d₁**) under the optimized reaction conditions was carried out to obtain the desired product **7aa** in 77% yield with 93% ee, where 30% of the acetylenic hydrogen atom of the product was found to be deuterium-labeled (Fig. 4d), suggesting that a proton transfer occurs for the presumable intermediary alkynyl complex with the deuterated pyridinium cation obtained via the cleavage of **4a-d₁** to afford alkyl radical. Here, the lower D content of the acetylenic proton of **7aa** may be due to the presumable H–D isotope scrambling among the deuterated pyridinium cation derived from **4a-d₁**, NH₄BF₄, and H₂O formed by the dehydration process from **3a**.” and Fig. 4d of the original manuscript in the revised manuscript. Thank you for your valuable suggestion.

- (5) Reviewer 3 pointed out "It seems the calculations in Figure 5 were performed with secondary propargylic alcohols without a CF₃ group. Could it be that the mechanism is different when a tertiary, CF₃ substituted propargylic alcohol (the actual substrate) is employed? On the very bottom right, a “TEMPO trapped” substrate is labeled as “not observed”, however, the authors would have observed the product originating from the tertiary propargylic alcohol actually employed in the reaction. I am not sure if the two pathways in 5b can be distinguished by a TEMPO experiment. The lack of formation of the homo-coupled product may not exclude the pathway in Scheme 5b, because it may just be more difficult to homodimerize two tertiary carbon centers. In reference 87, secondary propargylic alcohols are employed, and a comparison with the mechanism in the current manuscript is, thus, shaky." As pointed out by Reviewer 3 pointed out, we carried out the DFT calculations for the reaction of a secondary propargylic alcohol bearing a phenyl moiety at the propargylic position **1** with the benzyl radical shown in Fig. 8a of the revised manuscript (Fig. 5a of the original manuscript). As to the DFT calculations on the propargylic substitution reactions with nucleophiles, we already reported the mechanistic studies on the propargylic substitution reactions of not only the secondary propargylic alcohol **1** but also the tertiary propargylic alcohol containing a CF₃ moiety **3a** with a nucleophile (acetone) (ref. 33 of the revised manuscript (ref. 65 of the original manuscript): *Chem. Asian J.* **16**, 3514 (2021) and ref. 34 of the revised manuscript (ref. 66 of the original manuscript): *ACS Omega* **41**, 36634 (2022), respectively). According to results of the DFT calculations, the reaction pathway using the secondary propargylic alcohol **1** is almost the same with that using the tertiary propargylic alcohol **3a**. Based on our previous mechanistic studies on the propargylic substitution reactions with nucleophiles, we consider that the reaction of the tertiary propargylic alcohol **3a** with the benzyl radical proceeds via the similar reaction pathway to the reaction of the secondary propargylic alcohol **1** with the benzyl radical as shown in the present manuscript. On the other hand, Reviewer 3 also pointed that the two reaction pathways as shown in Fig. 8b of the revised manuscript (Fig. 5b of the original manuscript) cannot be distinguished by a TEMPO experiment. As pointed out by Reviewer 3, we cannot exclude the other possibility of the reaction pathway shown in Fig. 8b of the revised manuscript (Fig. 5b of the original manuscript). As a result, we have modified the descriptions in the second paragraph, page 27 of the original manuscript: “The corresponding propargylic radicals were also not trapped by TEMPO (Fig. 4c), ruling out the possibility of the formation of ruthenium(II) radical species before the attack of benzyl radical.” as “The corresponding

propargylic radicals were also not trapped by TEMPO (Fig. 8b), which may support the reaction pathway via the attack of benzyl radical at first, followed by the one-electron reduction. However, we cannot exclude the possibility of the other reaction pathway only by the TEMPO experimental result.” in the second paragraph, page 16 of the revised manuscript. Thank you for your valuable suggestion.

- (6) Reviewer 3 pointed out "Figure 1a (i) and (ii), "allyic positions" misspelled. Page 14, "syntetic" and "congainng" misspelled. Page 15, "allenylinde" misspelled. Page 8, "propagylic" misspelled. Page 22, "leadgin" misspelled." According to the suggestion, we have corrected all of the original descriptions in the revised manuscript. Thank you very much for your careful reading.
- (7) Reviewer 3 pointed out "In some references (like #12) the phrase "et al." is used, however, I think all authors should be listed." We agree with the comments by Reviewer 3 in other journals. However, we should follow the citation styles for *Nature* journals, where et al. is used for references with more than five authors. Thank you for your valuable suggestion.

REVIEWERS' COMMENTS

Reviewer #1 (Remarks to the Author):

My concerns have been addressed in the revised manuscript. I have no further comments. Although the substrate scope is still not very large, the author did something unique for CF3. I support the publication of this work.

Reviewer #3 (Remarks to the Author):

The authors revised the manuscript along the lines I suggested. The authors addressed the points I was raising and revised the manuscript along the lines I was suggesting. The authors revised their original idea of the formation of radicals and understood that the deuterium labeling experiment was not very meaningful. I am still not fully convinced about the stable allenylidene complex serving as an "intermediate", however, it may be that the allenylidenes are "semi-stable" without a nucleophile present. This point can be addressed further by scientific discourse. As such, I do not have any further comments to make.

The last paragraph "Discussion" should be labeled "Conclusion", I think.

Point-by-Point Response to the Referees.

As for the comments by Reviewer 1

Reviewer 1 pointed out "My concerns have been addressed in the revised manuscript. I have no further comments. Although the substrate scope is still not very large, the author did so something unique for CF₃. I support the publication of this work."

As shown here, Referee 1 recommended the publication of our previous version without further revision. Thank you very much for the recommendation for the publication in *Nature Communications*.

As for the comments by Reviewer 3

Reviewer 3 pointed out "The authors revised the manuscript along the lines I suggested. The authors addressed the points I was raising and revised the manuscript along the lines I was suggesting. The authors revised their original idea of the formation of radicals and understood that the deuterium labeling experiment was not very meaningful. I am still not fully convinced about the stable allenylidene complex serving as an "intermediate", however, it may be that the allenylidenes are "semi-stable" without a nucleophile present. This point can be addressed further by scientific discourse. As such, I do not have any further comments to make. "

As shown here, Referee 3 recommended the publication of our previous version almost without further revision. We believe that the present descriptions provide useful and enough information to readers of the present manuscript. Thank you very much for the recommendation for the publication in *Nature Communications*.

Reviewer 3 also pointed out " The last paragraph "Discussion" should be labeled "Conclusion", I think."

However, as pointed by you, we will not revise this label.